



**Winter mixing, mesoscale eddies and eastern boundary current: Engines for biogeochemical**
**variability of the central Red Sea during winter/early spring period.**
Nikolaos D. Zarokanellos[1] and Burton H. Jones[1]
[1]King Abdullah University of Science and Technology (KAUST), Red Sea Research Center (RSRC),
Biological and Environmental Sciences & Engineering Division (BESE), Bioscience program, Thuwal,
23955-6900, Saudi Arabia.
Correspondence to: Nikolaos D. Zarokanellos (nikolaos.zarokanellos@kaust.edu.sa)
**Abstract**
The central Red Sea (CRS) has been shown to be characterized by significant eddy activity throughout
the year. Weakened stratification in winter may lead to enhanced vertical exchange contributing to
physical and biogeochemical processes. In the winter of 2014-2015, we began an extended glider time
series to monitor the CRS where eddy activity is significant. Remote sensing and glider observations
that include temperature, salinity, oxygen, carbon dissolved organic matter (CDOM), chlorophyll
fluorescence (CHL) and multi-wavelength optical backscatter have been used to characterize the effects
of winter mixing, eddy activity and lateral advection. During winter and early spring, mixing up to 90m
driven by surface cooling and strong winds combined with eddy features was insufficient to penetrate
the nutricline and supply nutrients into the upper layer. However, the mixing events did disperse the
phytoplankton from the deep chlorophyll maximum throughout the upper mixed layer (ML) increasing
the chlorophyll signature detected by ocean colour imagery. In early spring, the eastern boundary
current (EBC) is evident in CRS. The EBC brings relative high concentrations of CHL and CDOM
along with lower oxygen concentrations indicative of previous nutrient availability. In addition to the
vertical mixing, mesoscale eddy activity cause 160m vertical displacement of the 180 μM isopleth of
oxygen, proxy of the nutricline interface. Within the cyclonic feature, this oxygen isopleth shallowed to
60m, well within the euphotic layer. Remote sensing analyses indicate that these eddies also contribute



to significant horizontal dispersion, including the exchange between the open sea and coastal coral reef
ecosystems. When the phytoplankton is distributed through the mixed layer clear diel variability was
evident in CHL concentration. The biogeochemical responses provide a sensitive indicator of the
mixing and eddy processes that may not be detectable via remote sensing. Sustained in situ autonomous
observations were essential to understand these processes.
**1. Introduction**
The Red Sea is a narrow and long basin surrounded by hot deserts, and is isolated from the
world's oceans with no major connection to water bodies other than the Indian Ocean via Bab el
Mandeb, which is located between the Asian and African continents. The Red Sea is characterized by
negligible fresh water input from rainfall and terrestrial runoff, high temperatures and high salinity due
to large evaporation rates (Edwards, 1987; Smeed, 1997; Sofianos et al., 2002) and is a predominantly
oligotrophic sea (Grasshoff, 1969; Edwards, 1987). Furthermore, the Red Sea contains one of the most
diverse marine ecosystems in the world with important economic and environmental impacts primarily
due to coral reefs (Belkin, 2009; Longhurst, 2007; Raitsos et al., 2013). Coral reefs are often referred to
as "marine rainforests" and have concentrated in them the most diverse life in the oceans, although they
are fragile and vulnerable to oceanic warming (Cantin et al., 2010).
The wind and thermohaline forces drive the large-scale circulation of the Red Sea (Patzert,
1974; Phillips, 1966). Until recently, in situ observations of the Red Sea have been limited (Bower et
al., 2013; Morcos, 1970; Qurban et al., 2014; S.Morcos, 1974; Sofianos and Johns, 2007; Zhai, 2014;
Zarokanellos et al., 2017a, Zarokanellos e al., 2017b). Our knowledge regarding the variability of the
basin–scale circulation relied primarily on ocean models (Clifford et al., 1997; Sofianos and Johns,
2002, 2003; Tragou and Garrett, 1997; Yao et al., 2014a, 2014b). Numerical simulations and remote
sensing studies indicate that much of the time, mesoscale eddies that can propagate both zonally and
meridionally prevail over most of the basin (Raitsos et al., 2013; Yao et al., 2014a, 2014b; Zhan, 2013).
Chen et al. (2014) showed that mesoscale eddies play a significant role in the transport of heat, salt and
biological and chemical constituents in the Red Sea. Also, they provide valuable ecosystem services for
the reef system, pelagic species, and marine mammals (Shulzitski et al., 2016). Although mesoscale





processes seem to have a substantial effect on the Red Sea ecosystem, in situ observations to study the interaction between eddies and the coastal region, and their role in the modulation of phytoplankton productivity and biomass within the Red Sea remain poorly understood.

In the parts of oceans like the CRS where vertical exchange is limited, mesoscale features are likely to play a fundamental role in controlling phytoplankton via local nutrient fluxes (Longhurst, 2007). The eddy activity influences both optical and biological properties in the oceans (Kheireddine et al., 2017; Mahadevan, 2014; McGillicuddy et al., 1998; Pearman et al., 2017; Siegel et al., 2008). Eddies also have an important impact on the vertical transport of nutrients in the upper layer of the water column (Martin and Richards, 2001; McGillicuddy et al., 2003). Eddy transport affects the biochemical variability of the CRS (Zarokanellos et al., 2017b). A variety of environmental forcing factors can also influence the biogeochemical fluxes in the upper layer of the Red Sea, including seasonal mixing, eddy interactions, and cross-shelf eddy interaction. Until now, only remote sensing studies have been done to examine the seasonal variability of phytoplankton (Acker et al., 2008; Racault et al., 2015; Raitsos et al., 2013). Despite the fact that remote sensing technology has enormous spatiotemporal coverage to sample the surface expression of the ocean, the persistent clouds and atmospheric aerosol, sun-glint, and sensor saturation over sand limit data acquisition in the Red Sea (Racault et al., 2015). The Red Sea remains underexplored despite the ecological impact on several bio-physical processes, including the basic circulation pattern and the water mass formation, which are still obscure and highly debated (Sofianos and Johns, 2007). However, satellite imagery suggests that there is an ecological connection between the coastal zone and basin. This is significant, especially for the coral communities that live at the basin edge (Acker et al., 2008). The satellite imagery of surface CHL often gives us evidence of the water exchange but still needed to confirm from in-situ observations. Conversely, satellite remote sensing has failed to inform us regarding the vertical structure of CHL, as ocean color measurements are limited to the first optical depth.

The Red Sea is experiencing increased anthropogenic pressure due to oil exploration, aquaculture, urban and industrial development, and proposed deep-sea mining. One aspect of the ecological response to the increased environment stress is the changes in the biological productivity due to the stressors; it is essential to undertake the long-term monitoring of carbon cycling. Primary





production is not only the base for the marine food chain, but it also has a significant role in
biogeochemical fluxes of carbon (Green et al., 2014). Using autonomous underwater vehicles (AUVs),
we are able to monitor the biogeochemical properties for different spatiotemporal scales, helping us to
improve our knowledge and reduce uncertainties associated with carbon budgets (Cetinić et al., 2012).
Eddies and fronts are often enriched with nutrients in the upper layer, enhancing the primary production
and increasing the particle concentration (Li et al., 2012; Ohman et al., 2012; Stramma et al., 2013).
There are many ways to understand the interaction between the physical and bio-optical properties of
the Red Sea. In the past, few studies have been performed in the Red Sea, and these have primarily
focused on surface optical properties (Brewin et al., 2015).
The optical properties should covary with the CHL concentration (Siegel et al., 2005), but other
constituents are capable of absorbing and scattering light, such as detrital particles, CDOM, and
suspended solids, which all contribute independently to ocean color. The present study aims to address
the influence of wind mixing, eddies, fronts, and lateral advection on the optical properties.
Furthermore, this study examines how these physical drivers modulate the optical properties in the
surface and subsurface of the CRS during the winter period. Additionally, relationships between CHL,
CDOM, backscatter at 532nm ($bbp_{532}$), and particular organic carbon (POC) have been examined and
provide information on the primary production and optical variability in time and space. This study also
aims to understand the role of eddies and fronts in carbon export via both gravitational sinking and
subduction.
The present study is organized as follows: in situ and satellite observations with methods
described in section 2. Section 3 addresses the atmospheric forcing and the key physical dynamical
responses in the CRS during the winter/early spring period. It also presents the spatiotemporal
variability of individual measurements and their relationship, and it describes how wind winter mixing,
mesoscales eddies and the EBC affect key optical parameters like CHL, CDOM, $bbp_{532,}$ and the derived
POC (herefor POC) during the winter period. Discussion and conclusions are provided in section 4 and
5, respectively.

**2. Data Sources and Methodology**

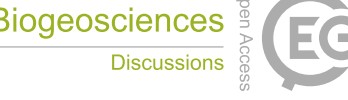

## 2.1 Glider Observations

Data from two Seaglider missions was obtained as part of the on-going research in the CRS (Fig 1a). The glider was deployed along an L-shaped transect, taking measurements for a total of 91 days during the winter period from December 2, 2014 to March 2, 2015 (Fig 1b & Table 1). The 250km L-shaped pattern was targeted toward to the center of a recurrent eddy that was indicated from remote sensing and in situ studies (Raitsos et al. 2013; Zhan et al. 2014; Zarokanellos et al. 2017a). Each deployment began offshore of the reef system Shi'b Nazar, with the glider piloted to offshore and returning to the inshore end of the Yanbu line. The glider completed the round trip in about three weeks. Each Seaglider measures a suite of parameters that include temperature and salinity (Sea-Bird Electronics' custom CT sensor with the unpumped conductivity cell), dissolved oxygen (oxygen optode 4831 sensor, Aanderaa), CHL, CDOM and phycocyanin (FL3-IRB sensor, WETLAB), and the customize backscatter at the following three wavelengths (532, 650 and 880; BB3 IRB sensor, WETLAB). For optical backscatter we present only measurements of 532 nm, as the overall patterns for 650 nm and 880 nm wavelengths were very similar. Potential temperature (hereafter, pot. temperature) by reference the measure temperature at depth to pressure at the surface . Both Seagliders have been programmed to profile from the surface to 1000 m at the nominal vertical speed of 0.18±0.02 m/s and moved horizontally at approximately 20-24 km day$^{-1}$. Each dive cycle took about 3-5 hours depending on the operation target depth, environmental conditions, and engineering control settings. However, for this study we only consider the upper 400 m layer of the water column. Spikes from the raw measurements have been removed from both hydrographical and biochemical parameters. In addition, CTD data has been examined for the thermal lag effect and no evidence has been observed.

The pre- and post-calibration activities were carried out at the Coastal and Marine Resources Core Lab (CMOR) glider facility at King Abdullah University of Science and Technology (KAUST) and included all ballasting, adjustment operations, and dark counts of the optical sensors necessary before and after the glider deployment. In addition, a comparison was made with CTD cast from hydrographical expedition in the study area, with the assumption that below 500 meters the water mass changes slowly relative to the time scale of sampling such that the T-S plots from both modes of sampling should tightly overlay each other (Zarokanellos et al., 2017a).



Evaporation rates within the Red Sea are of the order of 2 m year[-1] (Sofianos and Johns, 2002).
The high evaporation rates contribute to high salinity and significant affect the upper layer mixing (Lee
et al., 2000). To estimate the mixed layer depth (MLD) we used the criteria of 0.03 kg m[-3] where both
salinity and pot. temperature are consider by using the density criterion (de Boyer Montégut, 2004). The
MLD was calculated using the 1 m binned vertical profile data. CHL concentrations values based on the
factory calibrations were corrected by divided that number by 2. This correction is based on comparison
of fluorometer and extracted CHL measurements. The euphotic depth (Zeu) has been estimated by using
the vertical profiles of CHL and defined as the depth where Photosynthetically Active Radiation (PAR)
is reduced to 1% of its surface value (Morel and Maritorena, 2001). Glider fluorescence counts and
optical backscatter values obtained from the WET Labs Inc. FL3 and BB3 sensors were manufactured
calibrated. Raw scattering counts of optical backscatter minus dark counts were converted to total
volume scattering β (124,700 nm) using a factory-calibrated scale factor. The data from sensors were
converted to the engineering units by using the manufacturer calibrations:

$[\beta(\theta_c), \text{Chla}, \text{CDOM}] = \text{scale factor} \times (\text{output} - \text{dark counts})$          (1)

$\text{Scale Factor} = \frac{\alpha}{(\text{output}_\alpha - \text{dark counts})}$          (2)

$\text{Scale Factor} = E(\frac{b_b(532)}{\text{dark counts}}) \times T(\beta(\theta_c)/b_b(532))$          (3)

where scale factor (SF) from (2) and (3) is used in (1) for the computation of CHL (mg m[-3]), CDOM
(ppb) and total volume scattering at 532nm ($\beta(\theta c)$; m[-1] sr[-1]), respectively. In (2), the known
concentration a, is the constituent of CHL or CDOM and output$_\alpha$ is the measured signal output for the
known concentration. Furthermore, in equations (3) is the total backscattering coefficient (m[-1]), dark
counts is the signal of the sensor in the absence of light, which has been measured with a black tape
over the detector, and output is the measured signal output during field sampling. In both glider
missions, the optical face of the FL3 and BB3 sensor appeared clean with little to no fouling after each





deployment. The latter probably happened during the brief time that both gliders were in the euphotic

zone (<6%) and have been deployed in the most oligotrophic area of Red Sea. E and T in (3) are used

for denoting experimental and theoretical fractions, respectively. From β(θc), it is possible to obtain the

volume scattering of particles (βp) by subtracting the volume scattering of water (βw) (Zhang et al.,

2009). Then βp is used to compute the particulate backscattering coefficients (bbp; m$^{-1}$) and total

backscattering coefficients (bb; m$^{-1}$) following Boss and Pegau (Boss and Pegau, 2001). The bbp ($\lambda$) is

referred to simply as backscatter and can be expressed as:

$$b_{bp}(532) = 2\pi\chi\beta_p(532) \tag{4}$$

Temperature and salinity correction is performed on the sensor data to obtain the more reliable values of

bbp (Zhang et al., 2009). A detailed description of these optical computations can be found on the

official WETLABS website (http://wetlabs.com/eco-puck). Additional information about the

manufacturer's calibration of ECO Pucks' fluorometric measurements is found in Cetinic´ et al. (2009).

Total volume scattering converted to particulate volume scattering coefficients, βp, by subtracting the

volume scattering of seawater, βw (Morel, 1974), and then converted to particulate backscattering bbp,

by a factor of 2$\pi\chi$, where $\chi$ is 1.1 (Boss and Pegau, 2001). The optical backscatter at 532 nm has been

converted to 700 nm and then to POC concentrations using the various empirical relationships (Boss et

al., 2013; Cetinić et al., 2012; Tiwari and Shanmugam, 2013).

Regions with high irradiance like the Red Sea tend to experience chlorophyll fluorescence

quenching during the daylight hours (Behrenfeld et al., 1999; Kinkade et al., 1999). The fluorescence

quenching during the daytime tends to occur near the surface, but this can affect the florescence in the

upper layer of the water column, especially in waters with low $K_d$. Seasonal variability of

photochemical quenching has been observed, particularly during the spring to summer transition when

solar radiation is increasing, with the intensity of fluorescence quenching reduced at greater depth.

Correction of CHL concentrations where needed were computed from fluorescence that exhibited the

daily quenching near the surface (Sackmann et al., 2008; Xing et al., 2012), using as a reference the

MLD.



Because there is a linear relationship between the apparent oxygen utilization (AOU) and nitrate concentration we use AOU to estimate available nitrate in the water column (Naqvi et al., 1986). Recent evaluations of the relationship between AOU and nitrate plus nitrite (NOx) have validated the relationship describe by Naqvi et al. (1986) (Churchill et al., 2014; Zarokanellos et al., 2017b). Because the regression intercepts the oxygen axis between 180-185 µM, we have used the 180 µM oxygen isopleth as an index of the nitracline depth to examine the nutrient availability in the euphotic layer.

## 2.2 Meteorological data

The upper circulation of the CRS is strongly affected by the monsoonal reversal as the wind convergence occurring in this area during winter gradually changes to a more uniform pattern of northwest winds during summer (Zarokanellos et al 2017a). Although northwest winds prevailed over our study throughout the year (Papadopoulos et al., 2013), it is unclear how air temperature, the wind stress, and relative humidity affect the MLD in the Red Sea. Observation data sets from the meteorological station of Yanbu airport, which is located approximately 12 km east of the glider transect end point, have been used to investigate the relationship between the atmospheric forcing and MLD in the CRS (Fig. 1b). Meteorological data, particularly 6 hour wind speed, air temperature, and relative humidity, were obtained from the website (http://rp5.md/Weather_archive_in_Yanbu_%28airport%29). Wind stress was calculated from wind speed using the method of Large and Pond (1981). Wind speed, air temperature, and humidity were acquired at 10 m height, and daily mean values span the period from December 2, 2014, to March 2, 2015.

## 2.3 Satellite data

To place the glider observations into the larger context of the CRS, ocean color images were examined for the glider deployment period. MODIS Level-3 data were obtained from the NASA Ocean Color Web (http://oceancolor.gsfc.nasa.gov/).

## 2.4 Statistical analysis



Pearson linear correlation coefficient was computed for 4 different wind speed ranges between
air temperature and relative humidity versus the daily mean MLD (Table 2; Sokal and Rohlf, 1969). To
exam the relationship between the atmospheric forcing (air temperature, relative humidity and wind
speed) with the integrated CHL, CDOM, $bbp_{532}$, and POC in the MLD Pearson correlation has been
applied (Table 4). The transect line has been separated in three subdivisions (AB, ABC, and CD) as
indicated in Fig. 1b, where mean subdivisions values for CHL, $bbp_{532}$, CDOM, POC, and POC:CHL
were calculated and have been binned $(0.03^{o})$ latitudinally and longitudinally, respectively, for the
mixed layer, as Table 5 indicated. In addition, mean values of CHL, $bbp_{532}$, CDOM, and POC were
calculated for the first optical depth (Raitsos et al., 2013) and for the upper 200 m.
Principal component analysis (PCA) was used to investigate the linear relationship between
variables. In order to standardize the data forwards the variables have different units and magnitudes,
the PCA was performed using the correlation matrix instead of the covariance matrix. The following
data were used in the PCA analysis: potential (Temp), salinity (Sal), density (Den), oxygen (Oxy),
CDOM, $bbp_{532,}$ and CHL. The data were divided in four depths categories to explore the vertical
distribution and the interactions of the physical and biochemical processes in the CRS (Table 3). We
partitioned data into the following depth bins (surface-to maximum depth, surface-20, 20-100 and 100-
200). Principal component analysis is used to explain the correlations between the variables for each
depth bin.

**3. Results**
**3.1 Atmospheric conditions in the central Red Sea during winter/early spring**
Model analyses of the Red Sea indicate the importance of the wind forcing in the Red Sea
circulation (Clifford et al., 1997; Patzert, 1974; Quadfasel D, 1993; Sofianos and Johns, 2003; Yao et
al., 2014a, 2014b). The location of the convergence zone between northerlies and southerlies in the
central Red Sea plays a crucial role in the forcing of Red Sea circulation and waves during the winter
when winds in the southern half of the Red Sea are southerly (Langodan et al., 2015). The atmospheric
data allow us to understand the important role of wind speed during the winter mixing period and how
this affects the deep chlorophyll maximum (DCM). The air temperature, wind stress and relative





humidity from the meteorological station reveals the arrival of cold, intense northerly winds (Fig. 2). The intense wind stress of more than 0.004 N m$^2$ is associated with low relative humidity and producing mixing of the upper layer. The sea temperature follows the similar distribution with the air temperature. However, a time lag has been observed between the response of air temperature and sea temperature (5m; Fig. 2a). Air temperature fluctuations have been observed during the winter/early spring period, with the daily mean air temperature reached the minimum at 11$^{th}$ of January (15.2 $^o$C) and the daily mean MLD the maximum depth (85m) as the Fig. 2a and 2d indicated. A correlation between wind stress and MLD has been observed as indicated in the Fig. 2b and 2d. To determine the dependence of the MLD on wind speed, air temperature and relative humidity the data were subdivided into four wind categories in order to separate the effects of the wind from air temperature and relative humidity. We categorize the wind into low wind speed (0-2 m s$^{-1}$), medium wind speed (2-4 m s$^{-1}$), higher wind speed (4-7 m s$^{-}$1) and the entire range (0-7 m s$^{-1}$; Table 2). The correlation coefficient r is significant ($p<0.05$) between air temperature and MLD for the first and third wind speed category, but the linear relationship is week (~0.3). Furthermore, the relative humidity and MLD correlation coefficient r is significant ($p<0.05$) for first, three and last wind speed category (Table 2). Although, strong (negative) linear relationship (<-0.7) has been observed only in the last category.

### 3.2 Physical and chemical variability.

Based on the physical processes that occurred in the study area during the winter/early spring period, three distinct periods; a) the first period is dominated by winter mixing, b) the second period is characterized by intensification of the eddy circulation, and c) the last period demonstrate the interaction between the EBC and the eddies. Winter mixing was observed between 2 December 2014 to 10 January 2015. The rapid increase of the MLD coincided with strong wind stress events in the CRS (Fig. 2b and 2d). The eddy intensification spanned the period from 11 January until 10 February 2015, when a CE and AE formed in the study area. During the existence of the eddy intensification, the relative humidity and air temperature gradually increased, and the MLD became progressively shallower. The EBC/ eddy interaction was discernible from 11 February until 2 March 2015. During this




time, the MLD progressively increased and two wind events accompanied by decrease air temperature
occurred.
Glider measurements show how the salinity, potential temperature, stratification (Brunt–Väisälä
Frequency; BVF), oxygen, and NOx structure in the upper 400 m evolved during the winter/early spring
period (Fig. 3). The in situ observations illustrate the importance of winter mixing, the
physical/chemical role of the eddies, and EBC/eddies interaction. During the winter-mixing phase, the
MLD was highly variable extending from 8 to 88.2 m (mean 34.8 m, STD=24.5 m). During both
deployments, the euphotic zone range from 83.5 m to 126.2 m (mean 104.4 m, STD=7 m).
The glider observations reveal that the mixing events homogenized the upper layer to as deep as
90 m and as a result stratification is weak ($N^2 \sim 0$) as shown in Fig. 3a-b, and 3h. This weak stratification
extended to the depth of the permanent pycnocline, located between the 26 and 27.5 kg m$^{-3}$ $\sigma_\Theta$ (Fig. 3c).
The intense winter mixing thus oxygenates the upper layer to the depth of the mixed layer. During this
period, a small cyclonic eddy (CE) developed at the end of December 2014. The effect of the eddy was
to elevate the pycnocline reducing the MLD and lifting lower oxygen water into the upper layer (Fig.
3d). As with other variables the elevation of pycnocline brought nutrient containing water nearer to the
surface (Fig. 3e).
After the winter mixing period, the CE and AE eddies appears to intensify in the study area (Fig.
3f and 3g). Temperature decreased ($\Delta\theta \sim$ -2 ˚C) and salinity ($\Delta S \sim$ 0.3) increased rapidly in the upper
100m in a period of about 10 days (not shown here). The undulations of the pycnocline and isopleths of
other variables reflect the influence of the eddies on the hydrography of the region (Fig. 3f-j). In the
upper 100 m, the core of the CE is saltier (>40.2) and cooler (23.5 ˚C) than the surrounding water. In
contrast, the AE entrains warmer (> 25 ˚C) and less salty (39.8) water in its upper core. During the eddy
intensification period, the main pycnocline has significant weakens as indicated by the BVF (Fig. 3h).
Our observations show the development of the seasonal pycnocline at the end of January. The seasonal
pycnocline as indicated by the BVF follows the $\sigma_\Theta$ = 27 kg m$^{-3}$ isopycnal. The high BVF associated
with $\sigma_\Theta$ = 28 kg m$^{-3}$ isopycnal suggest that this density surface is good indicator of the permanent
pycnocline. Eddies during this period have an important role in modulate the vertical oxygen
distribution in the CRS (Fig. 3i). The highest oxygen concentrations (>200 μM) are observed at the core



of AE where the maximum concetrations penatrate to 90m. The thickness of the high oxygen layer
(>200 μM) decreased from 90m in the center of AE to less than 60m in the core of the CE. Based on
Naqvi et., al (1986) and Churchill et al., (2014) measurable NOx begins to appear at an AOU of about
10-11 μM. Thus, the top of nitricline lines up approximately with the $\sigma_\Theta = 28$ kg m$^{-3}$ isopycnal within
the eddies we see an elevation of this isopycnal from more 190m to less than 125m (Fig. 3i and 3j).
Within the AE, high oxygen concentrations (>180 μM) extend as deep as 200 m.

During EBC/eddies interaction period, lower salinity, warm water was found in the upper 100m

in the southern part and extended until 23.1 °N (Fig. 3k and 3l). The fresher, warmer water is consistent
with entrainment of water from the northward transport of Gulf of Aden water via the EBC as
previously described by Zarokanellos et al., (2017a and 2017b). During this period, the AE occupies the
south part of the study area. The AE entrains the EBC water of the center of AE and likely redirects the
EBC offshore around the periphery of AE (Fig. 3l). Zarokanellos et al. (2017b) observed similar
patterns in an earlier study in spring 2013. During EBC/eddies interaction period, the stratification
(BVF) increased in the upper 100 m (Fig. 3m). The presence of the EBC contributed to the increase of
stratification, as it advected warmer, fresher water into the region. Within the CE, the $\sigma_\Theta = 28$ kg m$^{-3}$
isopycnal rises from 220 m to 160 m in the core of the eddy. The CE lifts low oxygen, nutrient-rich
water toward the upper layer (Fig. 3n and 3o). An overall reduction of oxygen (<205 μM) occured in the
late winter/early spring (Fig. 3n). Very likely this reduction is associated with rising temperature
decreasing the oxygen saturation concentration. High values of the oxygen concentration (>205 μM) in
the surface layer were present in the core of the CE for the upper 60 m.

**3.3 Bio-optical variability.**

In the CRS generally, ocean color CHL is often a useful tracer to detect mesoscale eddies and to

understand of the biogeochemical processes associated with these features. Significant variations of
CHL, bbp$_{532}$, CHL to bbp$_{532}$ ratio, CDOM, and POC were observed along the glider track during the
winter/early spring period (Fig 4). Winter mixing and mesoscale eddy activity enhanced CHL
concentration in the ML during this period. Mean CHL concentration in the ML (0.118 ± 0.017) during
the observation period was higher then the integrated CHL (0.0984 ± 0.015; 0-200m). During the



mixing period, deep mixing distributed the phytoplankton through the mixed layer, increasing the signal
detected by ocean colour imagery but without a corresponding increase in integrated chlorophyll (0-200
m; Fig 4a). This temporal increase of chlorophyll concentration during the mixing period results from
entrainment of the deep chlorophyll maximum (DCM) into the mixed layer. These unstable conditions
of the upper layer (especially during the intense wind events) may obscure the biologically driven diel
cycle in the region. During the mixed period, relatively high concentrations of $bbp_{532}$ ($\sim10^{-3}$) have been
observed in the mixed layer and associated in several cases with high CHL concentration (Fig 4b). High
$CHL:bbp_{532}$ ratios have been observed primarily in the DCM and locally in the core of the CE where
CHL is elevated (Fig 4c). Moreover, a high $CHL:bbp_{532}$ ratio occurred after wind stress events with a
time lag of a few days. The wind stress events cause deepening of the mixed layer and increase
temporarily the available nutrients. During the mixed period, low (<0.5 ppb) CDOM concentration has
been identified in the MLD (Fig 4d). CDOM generally increases with depth, with the maximum
concentration appearing below 300 m. CHL concentration covary with POC as shown in Fig 4a and 4e.
Within the center of the CE eddy, the bulk of POC was located of the upper 60 m, where its
concentration was greater than 16 mg C $m^{-3}$.
Throughout the intensification of the eddy circulation period, rapid changes in the density are
related with the eddy structure in the region. Deepening of the mixed layer increases nutrient
availability resulting in increased CHL concentration. The highest CHL concentration occurred in the
center of the mesoscale CE eddy where the MLD shoaled from ~100 to 20 m (Fig. 4f). Subduction
occurred at the interface of the CE and AE eddy, resulting in higher CHL (>0.4 mg $m^-3$) and $bbp_{532}$
(>1.2x$10^{-3}$ $m^{-1}$) abundance at the boundary (Fig. 4f-g). High $CHL:bbp_{532}$ ratios have been observed in
the DCM at the interface of the CE and AE as indicated at Fig. 4h. Variation in CDOM distribution was
observed in the upper 200 m where the eddy pair occurred (Fig. 4i). Low concentration of CDOM (<0.5
ppb) was present primarily in the upper layer where $\sigma_{\Theta}$ was less than 27.5 kg $m^{-3}$. CDOM concentration
was elevated (depressed) in the center of the CE (AE). The overall POC concentration increased during
the intensification of the eddy period for the upper 100 m, and subduction of POC occurred at the
interface between the AE and CE (Fig. 4j). During this period, the highest POC concentration was
observed at the core of the CE reaching up to 24 mg C $m^{-3}$.

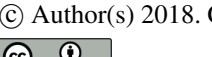



During the EBC/eddy interaction period, water with a high concentration of CHL (>0.4 mg m$^{-3}$)
was observed between 50 to 90 m, where the density anomaly was approximately 26 kg m$^{-3}$ (Fig. 4k).
Figure 4l shows that higher bbp$_{532}$ (>10$^{-3}$ m$^{-1}$) is associated with warmer and fresher water in the upper
70 m. bbp$_{532}$ was greater than 1.2x10$^{-3}$ m$^{-1}$ in the core of the CE. The CHL:bbp$_{532}$ ratio could was higher
in the DCM layer and at intermediate depths coinciding with the 26 kg m$^{-3}$ isopycnal (Fig 4m).
Zarokanellos et al. (2017b) used low salinity and elevated CDOM at intermediate depths to distinguish
Gulf of Aden Intermediate Water (GAIW) in the CRS. The water along the 26 kg m$^{-3}$ isopycnal where
both CHL and CDOM are elevated relative to the surrounding water is consistent with the GAIW
described by Zarokanellos et al. (2017b). GAIW was detectable from the southern end of the transect
until ~23 ºN, where the EBC and the CE intersected. Low concentration of CDOM (<0.5 ppb) is present
at intermediate depths in the center of the CE (Fig. 4n). Higher concentrations of POC (>30 mg C m$^{-3}$)
and CHL (0.4 mg m$^{-3}$) were detected in the core of the CE (Fig 4o). Through the EBC/eddy interaction
period, CHL and POC covaried (Fig. 4k and 4o). Subduction of CHL, bbp$_{532,}$ and hence POC is
indicated by the downward projection along the 27.5 kg m$^{-3}$ isopycnal southward from the eddy center
(Fig. 4k, 4i and 4o).

**3.4 Water masses characteristics.**
Water masses characteristics underwent significant transition during the winter/spring period.
The potential temperature-salinity (Θ/S) diagram from both glider missions is shown in Fig. 5, where
the separate panels are color-coded for time, CHL, CDOM, and oxygen concentration. The most
distinctive feature of the Θ/S distribution is the very abrupt transition from relative diffuse Θ/S
distribution in the upper layer prior mid-January to a tight, linear Θ/S pattern in the upper layer after
mid-January. This abrupt transition appears to be due to the intensification of the CE lifting the 27.5 kg
m$^{-3}$ isopycnal to near the surface and the arrival of low salinity water presumably from the Gulf of
Aden. The association with salinity range 39-39.4 psu, higher CHL (>0.3 mg m$^{-3}$) and CDOM (>0.9
ppb) for density anomalies between 26 and 26.5 kg m$^{-3}$ (Fig. 5b and 5c) is consistent with the
descriptions of GAIW provided by Churchill et al. (2014) and Zarokanellos et al. (2017b). GASW is
distinguished from GAIW, as it is warmer (>26˚C), less saline (<39 psu), lower in CDOM (<0.9 ppb)



and less dense ($\sigma_\Theta$ <26 kg m$^{-3}$). In this set of observations, oxygen does not show a significant difference between GASW and GAIW (Fig. 5d). Oxygen in the upper layer ranged from 170 to 200 (μM) between the 25.3 and 27.5 kg m$^{-3}$ $\sigma_\Theta$. The maximum oxygen concentrations (210 to 220 μM) in the surface water have been associated with the two strong wind events in the middle of December 2014 (Fig. 2b) that induced strong mixing that ventilated the surface layer. Above densities of 28 kg m$^{-3}$, oxygen concentration decreases from 150 to 20 (μM) at the depth of the oxygen minimum.

### 3.5 Physical and biological variability in time and space.

The CRS has been characterized as having high mesoscale activity and is one of the most oligotrophic areas of the Red Sea (Raitsos et al., 2013; Zhan et al., 2014). Until recently, the physical and biological variability have been examined with models, remote sensing studies, and infrequent in situ observations. This data set provides the first glimpse of the significant changes that can occur in a relatively short time period during the winter/early spring. Vertical profiles of the mean and standard deviation (STD) of salinity, pot. temperature, CHL, CDOM, bbp$_{532,}$ and POC illustrate the magnitude of the variability that was observed in this period (Fig. 6). The data was parsed into three time-periods based the vertical structure and spatial heterogeneity of the water masses in the upper layer. The bulk of the variability, particularly in optical variables, occurred in the upper 200m (Fig. 6). This is consistent with previous observations of eddy structure in CRS where evidence of the eddy penetrated to at least 200m (Zarokanellos et al., 2017a). Throughout the winter mixing period, the upper layer is homogenized of reducing the variability of the upper layer. During the eddy intensification period, we observed the greatest variability. This variability was likely due to seasonal transition and significant horizontal exchange through the interaction of complex eddy activity. The upper layer variability was again reduce when the EBC brought Gulf of Aden water that constrain the variability of the upper layer.

While remote sensing of ocean color data has provided significant insight into the temporal and spatial variability within the first optical depth (nominally ~ 20 m) of the Red Sea, the patterns might not necessarily represent the variability of the upper 200m. This is most evident in the comparison of the upper 20 m CHL versus the integrated CHL of 200 m (Fig. 7). In this case, the upper layer shows an increase in the near surface CHL of more than 50 %. Figure 7 shows box-and-whiskers plots providing



the mean values and standard deviations of the variables CHL, $bbp_{532}$ and CDOM for latitudinal (6
transects; 2 AB and 4 ABC) and longitudinal (4 transects CD) sections of both glider deployments. The
increase of the CHL in the first 20m results from deep mixing which entrains the DCM into the mixed
layer but without a net increase of the integrated water column CHL. This mixing is clearly evident in
the CHL distribution and MLD shown in Fig 4a. During the eddy intensification period CHL
concentration in the upper 20 m decrease as the DCM reformed but without significant change in the
integrated water column CHL (Fig 4a and Fig 7). However, along the line CD surface $bbp_{532,}$ integrated
CDOM and integrated $bbp_{532}$ increased during the eddy intensification phase. These increases are likely
due to enhanced along coast transport. During the EBC/eddy interaction period, both surface and
integrated $bbp_{532}$ increased along with integrated CHL. Despite the increase of integrated CHL the
surface CHL continuous to decrease (Fig. 7a-b). These results indicate that there is not always a tight
relationship between $bbp_{532}$ and CHL.

**3.6 Environmental factors controlling the optical properties.**

As described above, significant physical and optical variability occurred during the winter/early

spring period. To further characterize this variability principal component (henceforth PC) analysis was
used to determine the main modes of variability within the data. For the PC analysis, the data were
subdivided into four depth regions a) the upper 200m, b) surface layer to the first optical depth, c) DCM
region (20-100 m), and d) sub-euphotic region (100-200 m). Our aim is to reduce the effects of the
vertical variability of the physical and biological factors. The first two PCs accounted for 90.3% of the
variance for the upper 200m, 59% for the surface layer, 64.4% for the DCM layer, and 81% for the sub-
euphotic layer. Table 3 indicates the loading for the first two PCs and the relationships with respect of
each variable. For the upper 200m, PC1 explains 81.8 % of the total variance and was characterized by
negative loading in potential temperature, oxygen, CHL, and $bbp_{532,}$ and positive in salinity, density,
and CDOM (Table 3). The first PC generally accounts for the vertical structure of water column. For the
second PC, CHL, CDOM and oxygen show similar absolute magnitude but oxygen was in opposite in
sign to CHL and CDOM. Within the upper 100 m salinity along with oxygen concentration are opposite



in sign to CHL and CDOM. This pattern of loadings is consistence with the characteristics of GAIW
where lower salinity, lower oxygen GAIW is accompanied by higher CHL and CDOM concentrations.

For the surface layer (0-20m), the first component is dominated by the physical variables (T, S,

$\sigma_\theta$) where T is opposite in sign with S and $\sigma_\theta$. The second component characterize the bio-optical
variability where CHL, CDOM and $bbp_{532}$ are opposite in sign with oxygen. The first two components
that characterize the DCM follow a similar pattern with the upper 200m. PC1 was dominated by the
physical variables and oxygen concentration. Within PC1 temperature and oxygen are opposite in sign
with salinity and $\sigma_\theta$ similar with PC1 of the overall structure of the water column. The bio-optical
variables were more significant in PC2. In the sub-euphotic region, the PC1 again dominated by the
physical parameters. PC2 is dominated primarily by $bbp_{532}$.

## 4. Discussion.

Red Sea is a marginal sea affected by strong evaporation. When evaporation is accompanied by

significant cooling the water column stratification weakens such as strong surface forcing can cause
deep vertical mixing. If the mixing penetrates the nutricline, water column productivity may increase
due to the injection of nutrients in the euphotic zone. To investigate the effect of atmospheric forcing on
the biological distributions of CHL, CDOM, $bbp_{532}$, and POC of the ML, correlation coefficients were
calculated between characteristic atmospheric variables (air temperature, relative humidity, and wind
speed) and biological variables (CHL, CDOM, $bbp_{532}$, and POC) in the ML for both glider missions
(Table 4). The negative relationship of air temperature and relative humidity with CHL (-0.46 & -0.47)
and CDOM (-0.47 & -0.34) is consistent with weakened stratification resulting from cooler temperature
and increase salinity. The weakened stratification allows for mixing that penetrates the pycnocline
entraining deeper water that contains higher CDOM and nutrients concentrations. CHL and CDOM are
weakly correlated with the wind speed (0.32 & 0.30, respectively). No significant correlations exist
between wind speed with $bbp_{532}$ and POC.

Mean values of CHL, CDOM, $bbp_{532}$, POC, and POC:CHL in the mixed layer were calculated

separately for the two legs of the glider pattern (Table 5). The mean CHL concentration in the mixed
layer is 0.12 mg m$^{-3}$. CHL concentrations along segments ABC-1 to ABC-3 increased by nearly 50%



when deep mixing and mesoscale eddy activity were present. CHL increased along the segment CD-2 when an intrusion of GAIW occurred (Fig. 3b). Increase of CDOM concentration in the mixed layer has been observed under two processes during the winter. Mesoscale eddies are one process that results vertical transport of sub-pycnocline water that would contribute to increase CDOM in the upper layer.

Lateral advection of GAIW containing elevated CDOM can also contribute CDOM to region (GAIW; Zarokanellos et al 2017b). An overall latitudinal increase of CDOM has been revealed between transects ABC-2 to ABC-4 (Fig. 7f). Moreover, enhanced of CDOM has been also observed in the longitudinal transects between the CD-2 and CD-4. The overall $bbp_{532}$ concentration in MLD during the study period was 8.77E-04. GASW appears in CRS in the late winter/early spring and is correlated with elevated $bbp_{532}$, under the assumption that phytoplankton CHL decomposes faster than total organic C. POC:CHL ratio has been used to differentiate when or where autotrophic processes dominate (POC:CHL<200) versus heterotrophic processes (POC:CHL>200; Bentaleb, 1998; Cifuentes et al., 1988). Heterotrophic remineralization by microbial organisms is understood to dominated when the POC:CHL ratio is greater than 200-300 (Bentaleb, 1998; Cifuentes et al., 1988). Based on these assumptions, heterotrophic remineralization was important at the beginning of the winter period as shown on Table 5. After the first mixing event in December 2014, the contribution of autotrophic processes in the mixed layer increases based on the decrease of the POC:CHL ratio (<160).

The mechanisms that generate and sustain mesoscale eddies in the CRS are so far poorly understood. In the oligotrophic CRS, there is an inconsistency of nutrient supply and replacement in the euphotic zone. Vertical exchange between the sub-pycnocline region and the euphotic zone is essential for phytoplankton growth and productivity. Mesoscale eddies in the CRS have a large biogeochemical affect not only in the CRS but contribute to the larger-scale processes in the northern half of the Red Sea (Zarokanellos et al 2017a, b). Globally, oligotrophic waters can contribute more than 30% of the marine carbon fixation and efficiently transfer atmospheric $CO_2$ to the ocean interior via physical and biological pumps (Marañón et al., 2003; Moutin and Prieur, 2012). The export of organic carbon from the surface product layer to the deeper ocean is a critical component of the carbon cycle. Many studies until now, provide controversial results about the POC flux between cyclonic and anticyclonic eddies (see Table 1; Shih et al., 2015).





During the eddy intensification period, eddies have a key role in the exchange of water mass
characteristics and optical properties between the northern and the southern regions of the Red Sea
(Zarokanellos et al., 2017a). Glider observations and satellite chlorophyll have been used to characterize
not only the horizontal but also the vertical distributions of CHL. Figure 8 shows coincident in-situ and
remote sensing observations of CHL, $bbp_{532}$ and CDOM during 1 to 11 February, 2015. The in-situ
observations reveal subduction of CHL and suspending particles ($bbp_{532}$) at the interface of the eddy
pair and elevated concentration of CHL, and $bbp_{532}$ has been observed at the edge of the AE (Fig 8 and,
9b). Maximum integrated CHL concentrations occur at the interface between AE and CE features where
the subductive CHL of the CE and the shallower DCM of the AE overlap (22.9 ˚N; Fig. 8d). In the same
period, the EBC that brings low salinity and rich in nutrients water (GAIW) at intermediate depths in
the CRS. GAIW has been characterized by elevated CHL and CDOM concentrations and lower salinity
(Fig 8a, 8c and 3i). The arrival of the EBC advects both GAIW and GASW into the region where both
water masses contain high $bbp_{532}$ (Fig. 8b and 8c). The AE accumulate GASW and GAIW in the core of
the eddy and as result elevate normalize average and integrated CHL has been observed (Fig 9d). The
CE enhances the upward displacement of deep, nutrient rich water to the euphotic zone. The latter
contribute in the increase of CHL, $bbp_{532}$ and CDOM at the core of the CE, which located at ~23.3 ˚N.
Remote sensing analyses indicate that the eddy activity contributes to horizontal CHL
dispersion including the exchange between open sea and coastal environments. An ongoing question
has been whether the surface ocean color signal associated with eddies are truly chlorophyll signatures
or perhaps CDOM or particulate organic material that originates from coastal regions. Based on the
comparison of the in-situ glider observations with the remotely sensed ocean color in this example it
appears that the signature is generally chlorophyll that results from the interaction of the eddy with the
ambient ocean. Eddies spanning the basin width can interact with the coastal region entraining organic
matter from shallow coastal environments and transport that across the width of the basin (Fig. 8e and
8f). The biogeochemical response to the subsurface physical processes provides a sensitive indicator to
the processes that result from the mixing and eddy dynamics – processes that are not necessarily
detectable via remote sensing.



In general, the export of POC, estimated from bbp532, from the euphotic zone of the ocean is primarily considered to be due to sinking of phytoplankton, particulate aggregates, and fecal material (Durkin et al., 2016; Siegel et al., 2014). In this study, we show that the interface of the eddy pair subduction of particles from the surface layer (>=100m) transports them to depths of nearly 200m beneath the interface. This results is consistence with the growing body of literature that demonstrates the importance of eddies and fronts in the downward flux of organic matter beneath the euphotic zone (Ashjian et al., 2006; Jones, 1991; Kadko et al., 1991; Shih et al., 2015; Stukel et al., 2017; Washburn et al., 1991)

A fundamental question of understanding Red Sea is how Gulf of Aden Water interacts with the Red Sea. It is generally understood that GASW enters the Red Sea during September to May under the influence of the north-east monsoon (Murray and Johns, 1997; Smeed, 2004; Sofianos et al., 2002; Yao et al., 2014a) and GAIW enters during the period of the south-west monsoon when winds in the southern Red Sea are toward the south-southeast. GAIW is known to be low in oxygen, high inorganic nutrients and elevated in CHL (Churchill et al., 2014; Sofianos and Johns, 2007; Wafar et al., 2016; Zarokanellos et al., 2017a). Recent results show that GAIW also contains elevated CDOM fluorescence (Zarokanellos et al., 2017b). The advection of low salinity water into the study region appears from the TS perspective to be GASW that has enter during the winter period. However, embedded within the TS structure is a subsurface region where both CDOM and CHL concentration are elevated above the water with lower and higher densities along where appears to be a simple mixing line. A possible explanation of this could be there under strong wind forcing near the strait of Bab en Mandab flow of Gulf of Aden Water into the Red Sea may entrain some GAIW as well as GASW. Thus the winter injections of water of the Gulf of Aden may also enhance the nutrient flux and productivity of the Red Sea but not to the extent of the summer GAIW intrusions.

Vertical chlorophyll distributions can provide an indication of nutrient availability, and the role of intrusion from Gulf of Aden Intermediate Water (GAIW) in the CRS. GAIW is characterized by low salinity, high CDOM and nutrient concentration (Zarokanellos et al., 2017b). GAIW is the primary source of nutrients into the Red Sea (Naqvi et al., 1986; Souvermezoglou et al., 1989). However, its spatial distribution and its contribution to primary production within the basin remains unclear. Previous





studies have shown that GAIW intrusions are seasonal and episodic in character (Churchill et al., 2014; Sofianos and Johns, 2007; Zarokanellos et al., 2017b). GAIW intrusions affect both physical and biogeochemical properties of the water. Once it has entered the Red Sea, northward advection of GAIW modifies its physical and optical characteristics as the result of solar insolation, strong evaporation and mixing with ambient Red Sea water (Fig 3k-l, 3n, 4k, and 4n). Nutrients become diluted and depleted as the GAIW advects northward. High CHL concentrations in the DCM are not always dependent on the presence of GAIW, but also depend on eddy activity that can elevate subpycnocline nutrients into the photic layer. GAIW, typically located between the 26 and 26.5 kg m$^{-3}$ isopycnals has been characterized by high concentration of CDOM (>0.9 ppb) and CHL (>0.4 mg m$^{-3}$; (Zarokanellos et al., 2017b). During the EBC/eddy interactions both GASW and GAIW were detected in the study area below 23 ˚N. These two water masses differ not only in their thermohaline characteristics, but also show distinct optical signatures.

Figure 9a shows a positive relationship between CHL and bbp$_{532}$ where the data are color coded according to salinity. For salinities less than 39.4 psu and densities of 26 kg m$^{-3}$ or greater, GAIW can be distinguished from GASW, by CHL concentrations >0.4 mg m$^{-3}$.and CDOM concentration >0.9 ppb (not shown here; Zarokanellos et al., 2017b). Figure 9b illustrates the same relationship but color-coded into 3 depth categories: a) surface layer to 20 m (approximately the first optical depth); b) 20 m-100 m, incorporating the DCM region; and c) the upper 200m. High values of bbp$_{532}$ occurred in the first depth category (0-20m), during winter and early spring period. CHL concentrations were low in this layer. Because we do not see significant diel variability in the CHL layer we conclude that this CHL fluorescence is not due to quenching. The high bbp$_{532,}$ low CHL water is associated low salinities that was conclude that the water is GASW but we cannot characterize the nature of the particles in this dataset. In the DCM layer, CHL and bbp$_{532}$ are highly correlated. Vertical elevation of the nutricline, subduction of organic matter and lateral advection are some of the physical processes that affect the optical characteristics of GASW and GAIW during their northward transport along the eastern boundary of the Red Sea.

**5. Summary and conclusions**



The Red Sea's significant latitudinal range from 10˚N to 30˚N creates a thermohaline circulation
that has similarities to larger scale ocean basins such as the Atlantic Ocean. In addition to its large-scale
circulation, mesoscale circulation patterns are ubiquitous within the basin. The basin responds to
various large-scale climatological forcing processes such as ENSO and North Atlantic Oscillation.
Warming of the upper layer increases stratification, thus reducing vertical mixing and the flux of
nutrients into the upper layer. Consequently, this stratification limits primary production. Because of
these characteristics, the Red Sea provides a laboratory for understanding the influence of climate
change on the physics and biogeochemistry of the upper ocean. Despite its potential importance for
understanding this until recently, limited knowledge has been available regarding dynamics of the Red
Sea.
In-situ and remote sensing observations have been used in this study to investigate the response
of physical and biological processes to atmospheric forcing during the winter and early spring period.
Our findings provide new observations of the physical and biological responses in the CRS. The glider
observations reveal temporal and spatial responses to atmospheric forcing. The responses include
vertical mixing, mesoscale eddies and lateral advection that are otherwise difficult to resolve with more
traditional ship-based or moored observations. Within this study both anticyclonic and cyclonic were
apparent during the observational period.
During the winter period, surface cooling and strong winds caused mixing in the upper 100m.
The intense mixing affects the distribution of all properties, including bio-optical properties. The deep
mixing redistributed the phytoplankton from the DCM through the mixed layer increasing the
concentration in the upper water column thereby increasing the chlorophyll detected by MODIS.
Despite the increase in nearsurface chlorophyll, the integrated (0-200m) CHL remained essentially
constant. The strong mixing events created the pre-conditions for the development of a mesoscale CE at
the end of December 2015 (Fig 3a and 3b). The mesoscale CE elevated nutrient-bearing subsurface
water into the euphotic zone leading to an increase in CHL concentration (Fig 4a). High concentration
of $bbp_{532}$ and POC has were also observed in the center of mesoscale CE (Fig 4b and 4e). During the
eddy intensification period, weakening of stratification and shallowing of the MLD was observed (Fig
3h). The structure of the density field suggests both a CE and AE were present. Within the CE



integrated chlorophyll and estimated POC concentrations (0-200m) were higher where the isopycnals
were uplifted, suggesting a response to shallowing of the nutricline in central region of eddy (Fig. 4f
and 4j). In February, deepening of the DCM along with the isopycnals along the southern boundary of
the CE and the two chlorophyll maximum layers indicates subduction at the boundary between the two
features. At this interface where the two CHL layers occur, the integrated CHL and bbp$_{532}$
concentrations are higher than to either side of the interface. During EBC/eddy interaction period,
northward transport along the northeastern boundary of the eddy is consistent with the presence of the
EBC in the CRS. During this period, water that is identifiable with GASW and GAIW change modify
both thermohaline and optical characteristics. Throughout this period, glider observations show that the
main source of fresh and rich in nutrient water is present at intermediate depths where lateral advected
with the EBC. The interplay between the EBC and eddies, regulate the distribution of the GASW and
GAIW within the CRS.

Better understanding of the physical dynamics and ecological response of the region requires

continued long-term monitoring. In order to better understand the biogeochemical dynamics of the
region, further efforts require elucidation of the primary productivity response to seasonal, mesoscale
and longer term processes Autonomous underwater vehicles (AUVs) equipped with the appropriate
sensor suite enable monitoring of physical and biogeochemical properties over a range of temporal and
spatial scales, helping to achieve these goals. The direct and indirect effects of isopycnal displacement
on phytoplankton distribution and productivity, close alignment of physical and biological properties at
frontal boundaries, and variation of physical and biogeochemical features of eddies, underscore the
importance of high-resolution in situ measurements at appropriate scales. It is well known that eddies
and fronts contribute nutrients into the upper layer enhancing primary production, phytoplankton
biomass and particle concentration (Li et al., 2012; Ohman et al., 2012; Stramma et al., 2013). We
believe that continuing to study these processes in the globally extreme Red Sea will help us to better
understand the potential response of the global ocean to the current trends in climate.

**Acknowledgements**
The authors gratefully acknowledge the NASA Goddard Space Flight Center, Ocean Ecology
Laboratory, Ocean Biology Processing Group for remote sensing data used in this study. Datasets from
ocean color are freely accessible online on the official website (https://oceancolor.gsfc.nasa.gov/)
accessed on 18/09/2016. Glider data obtained from both glider missions can be obtained from Nikolaos
D Zarokanellos (KAUST) and Burton H. Jones (KAUST). The authors are grateful the KAUST Coastal
Marine Operation Lab (CMOR) for their engineering support during the glider deployments. Particular
thanks go to Sebastian Steinke, Brian Hession, Samer Mahmoud and Lloyd Smith for their help with the
glider deployments. Funding from King Abdullah University of Science and Technology (KAUST)
supported the research in this publication.

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





Table 1: Summary of measured parameters and associated sampling period for each glider mission

| Seaglider Sensors | Dives | Deployment Period |
|---|---|---|
| C, T, P, BB3, FL3, $O_2$ | 155 | 2-14 Dec 2014 |
| C, T, P, BB3, FL3, $O_2$ | 546 | 17 Dec 2014 – 2 Mar 2015 |


Table 2: Pearson correlations for air temperature and relative humidify versus mixed layer depth (MLD)
for four wind speed categories. Bold values indicate where $p < 0.05$, and where the correlation coefficient
$|r|$ is > 0.7.

| Wind Categories | Air temperature vs MLD | | Relative humitidy vs MLD | |
|---|---|---|---|---|
| | r | p | r | p |
| 1) Wind Speed (0-7) | 0.298 | **0.005** | 0.246 | **0.02** |
| 2) Wind speed (<= 2) | 0.375 | 0.152 | 0.325 | 0.219 |
| 3) Wind Speed (>2 & <=4) | 0.301 | **0.019** | 0.294 | **0.022** |
| 4) Wind speed (> 4 & <=7) | -0.084 | 0.805 | **-0.704** | **0.016** |


Table 3: Matrix with the loadings from the first two PCs. Bold number indicates the dominant
environmental parameters in each PC, indicated by the loadings with the highest absolute values. The
analysis of the winter period includes data for both glider missions, and for all fourth-depth ranges.

| Both missions | The upper 200m | | 0-20 m | | 20-100 m | | 100-200 m | |
|---|---|---|---|---|---|---|---|---|
| Variables | PC1 | PC2 | PC1 | PC2 | PC1 | PC2 | PC1 | PC2 |
| Temp | **-0.41** | -0.07 | **-0.58** | -0.06 | **-0.48** | -0.10 | **-0.43** | **-0.15** |
| Sal | 0.40 | -0.13 | **0.48** | -0.27 | 0.48 | -0.20 | 0.42 | 0.19 |
| Den | **0.41** | 0.01 | 0.64 | -0.03 | **0.53** | 0.00 | **0.44** | 0.19 |
| Oxy | -0.37 | **-0.48** | -0.14 | **-0.50** | -0.28 | **-0.38** | -0.41 | -0.13 |
| CHL | -0.35 | 0.41 | 0.00 | 0.54 | -0.19 | 0.58 | -0.35 | 0.21 |
| CDOM | 0.37 | 0.48 | 0.07 | **0.56** | 0.24 | **0.60** | 0.33 | -0.05 |
| bbp$_{532}$ | -0.33 | **0.59** | 0.01 | 0.27 | -0.30 | 0.33 | -0.21 | **0.92** |






Table 4: Pearson linear correlations between the atmospheric variables (air temperature, relative humidity and wind speed) versus mixed layer CHL, CDOM, bbp532 and POC during the study period (both gliders missions). Where p-value is small (p<0.05) then the correlation coefficient r is significant.

| Variables | Mean CHL in ML | | Mean CDOM in ML | | Mean bbp$_{532}$ in ML | | Mean POC in ML | |
|---|---|---|---|---|---|---|---|---|
| | r | p | r | p | **r** | **p** | **r** | **p** |
| Air temperature | -0.46 | <0.01 | -0.47 | <0.01 | 0.18 | <0.10 | 0.13 | <0.21 |
| Relative humidity | -0.41 | <0.01 | -0.34 | <0.01 | 0.05 | <0.66 | 0.01 | <0.94 |
| Wind speed | 0.32 | <0.01 | 0.30 | <0.01 | 0.01 | <0.94 | 0.02 | <0.85 |

Table 5: Mean mixed layer CHL, CDOM, bbp$_{532}$, POC and POC:CHL ratio for latitudinal and longitudinal binned ($0.03^{o}$) transect during the study period. Grey cells highlight the values that exceed 50%.

| | Date | CHL in ML | CDOM in ML | bbp$_{532}$ in ML | POC in ML | POC:CHL in ML |
|---|---|---|---|---|---|---|
| **M252-AB** | AB-1: 2014/12/02-2014/12/08 | 0.10 | 0.65 | $7.97*10^{-4}$ | 14.91 | 287 |
| | AB-2: 2014/12/08-2014/12/14 | 0.10 | 0.65 | $8.16*10^{-4}$ | 13.52 | 325 |
| **M250-ABC** | ABC-1: 2014/12/17-2014/12/26 | 0.15 | 0.73 | $8.10*10^{-4}$ | 12.97 | 55 |
| | ABC-2: 2015/01/03-2015/01/14 | 0.17 | 0.81 | $7.58*10^{-4}$ | 13.26 | 46 |
| | ABC-3: 2015/01/14-2015/01/23 | 0.12 | 0.78 | $7.80*10^{-4}$ | 13.19 | 76 |
| | ABC-4: 2015/02/04-2015/02/15 | 0.11 | 0.80 | $9.36*10^{-4}$ | 14.73 | 115 |
| **M250-CD** | CD-1: 2014/12/26-2014/12/30 | 0.10 | 0.70 | $8.11*10^{-4}$ | 14.18 | 95 |
| | CD-2: 2014/12/30-2015/01/03 | 0.13 | 0.75 | $7.82*10^{-4}$ | 13.87 | 60 |
| | CD-3: 2015/01/23- | 0.07 | 0.69 | $9.14*10^{-4}$ | 15.69 | 144 |



| | | | | | | |
|---|---|---|---|---|---|---|
| | 2015/01/27 | | | | | |
| | CD-4: 2015/01/28-2015/02/04 | 0.07 | 0.75 | $9.78*10^{-4}$ | 17.48 | 157 |
| **Both missions** | Mean | 0.12 | 0.75 | $8.77*10^{-4}$ | 14.96 | 122 |
| | STD | 0.02 | 0.09 | $1.29*10^{-4}$ | 3.19 | 99 |





**Figures**


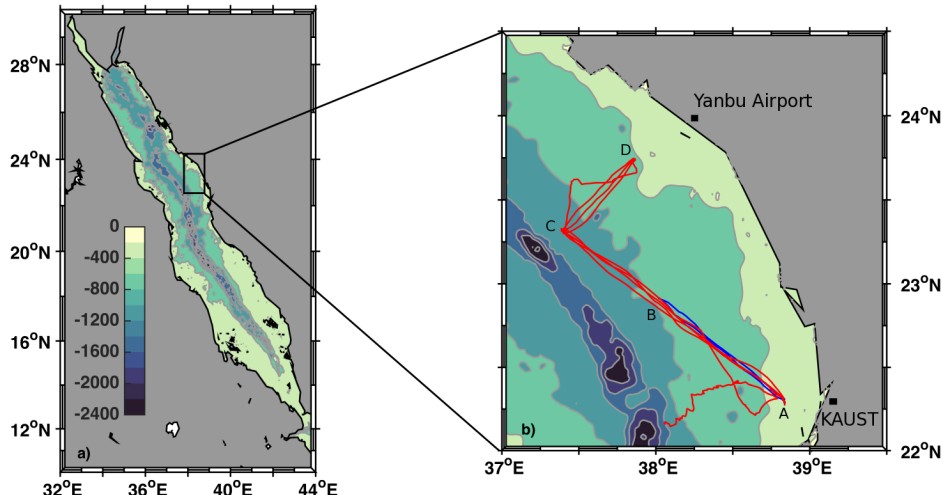


**Figure 1:** a) Bathymetric map of the Red Sea b) Bathymetric map of the study region with the two glider missions. The blue line represents the glider mission that was conducted between 02/12/2014 to 14/12/2014 and the red line represent the glider mission from 17/12/2014 to 02/03/2015. The letters A,B,C and D represent turning points of the glider missions segments.






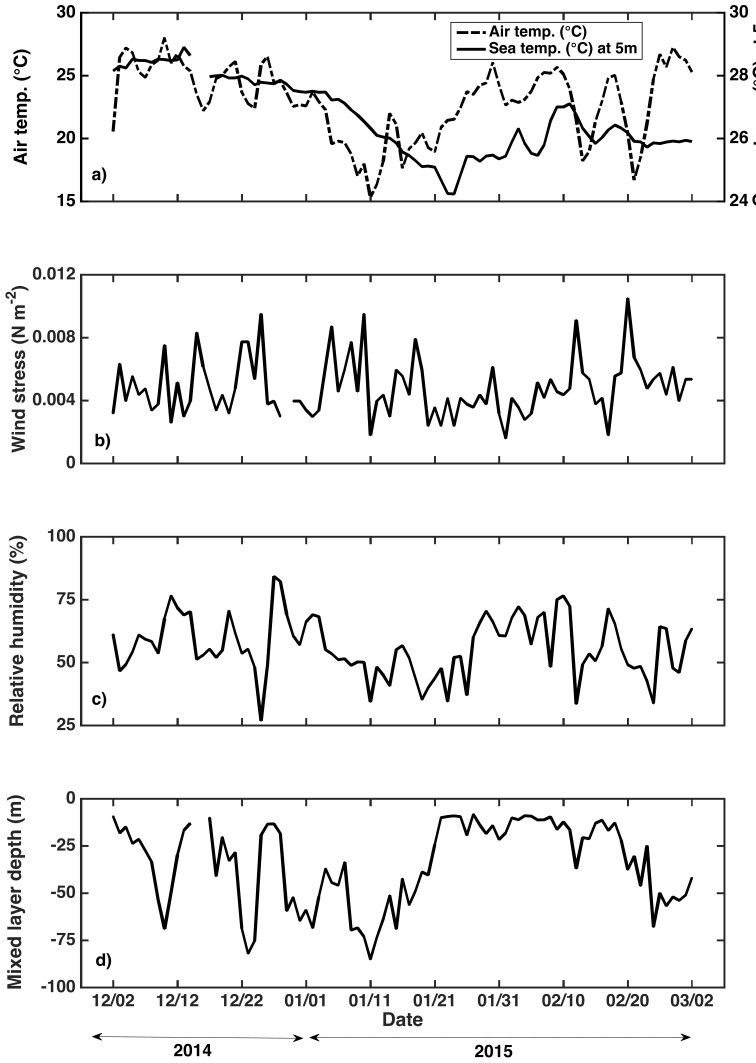


**Figure 2:** Daily mean of a) air temperature (°C) b) wind stress (m s⁻¹) c) relative humidity (%) and

Mixed Layer Depth (MLD) for the central Red Sea from December 2, 2014 to March 2, 2015.

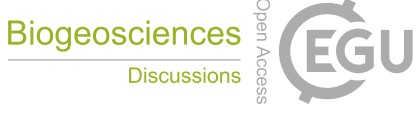


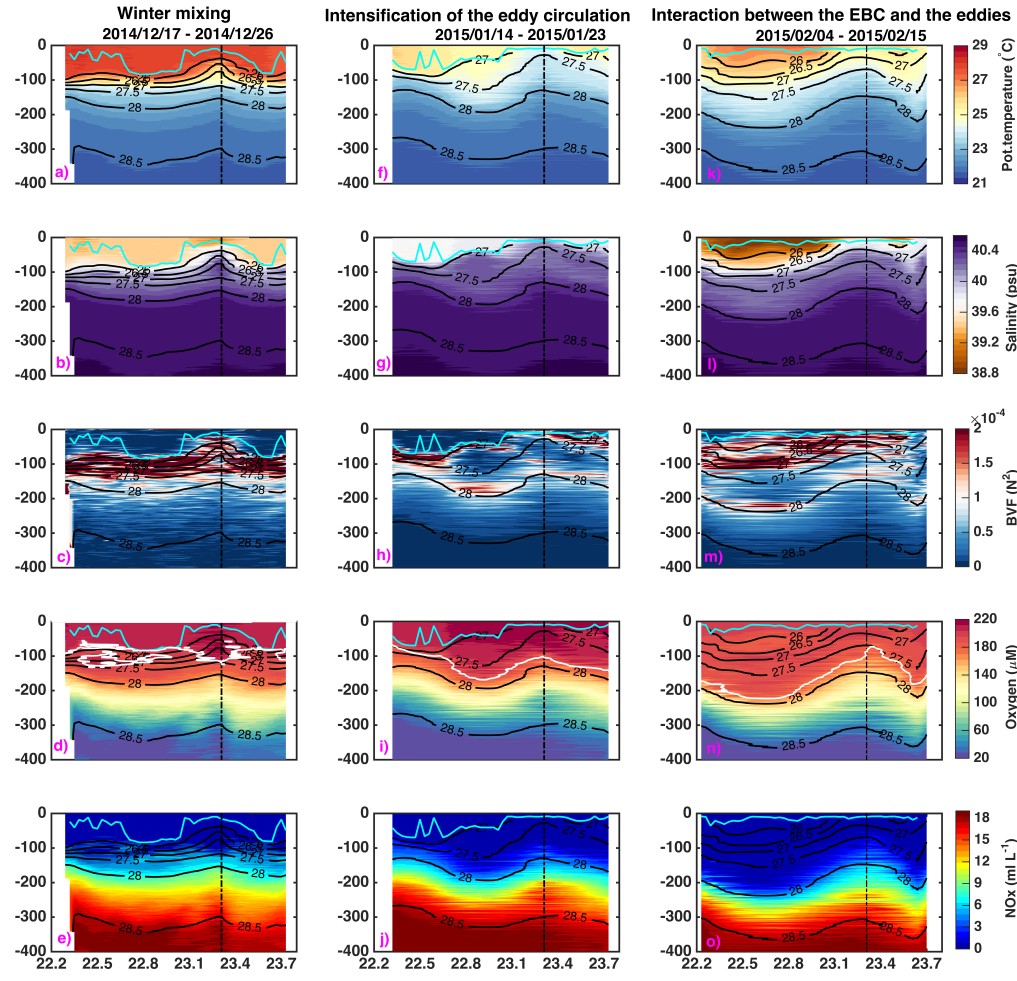



**Figure 3:** Seaglider sections plots for pot. temperature (˚C), salinity (psu), Brunt–Väisälä Frequency
(BVF; $N^2$), oxygen (µM), and NOx (ml L$^{-1}$) during the winter mixing period (a-e), the period of





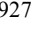

intensification of the eddy circulation (f-j) and the interaction between the EBC/eddy interaction period
(k-o). The vertical bold dash-dot black line indicates the turning point (C) as shown in Fig 1b.
Isopycnals are indicated with a solid black line (contour interval = 0.5 kg m$^{-3}$). The magenta line
represents the mixed layer depth (MLD) and the white line represents 180 (µM) isopleth.

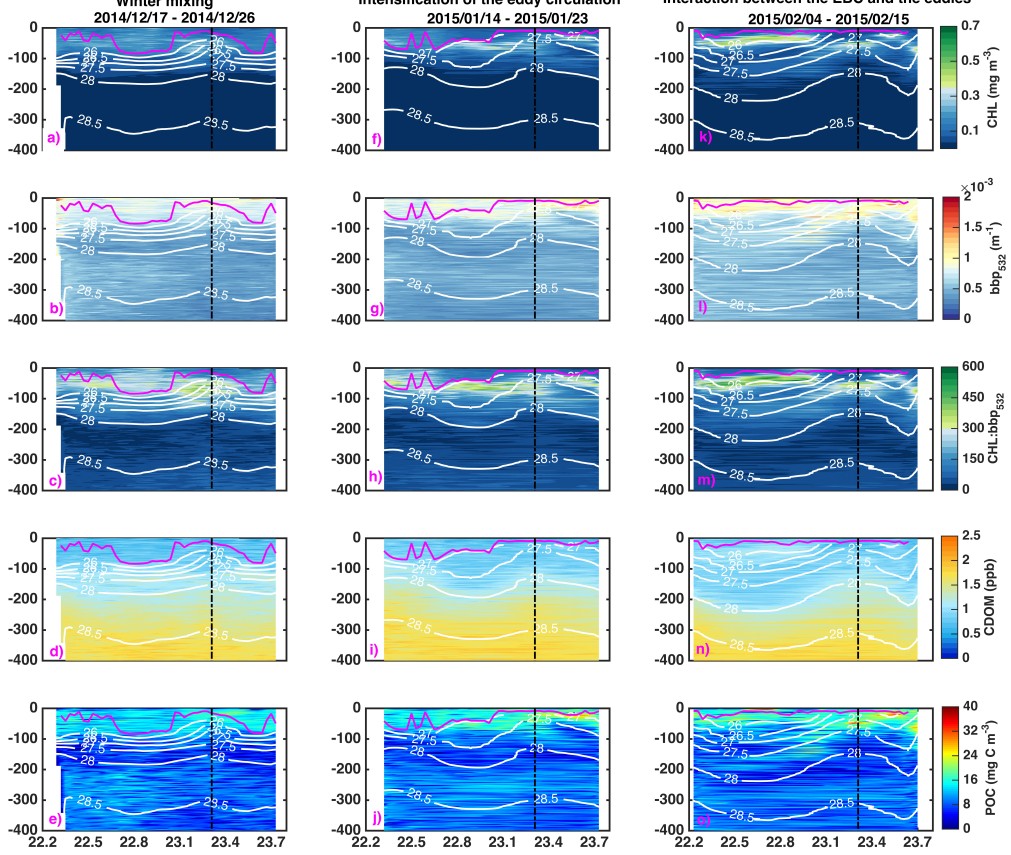


**Figure 4:** As in figure 3 but for CHL (mg m$^{-3}$), bbp$_{532}$ (m$^{-1}$), CHL:bbp$_{532}$, CDOM (ppb), POC (mg C m$^{-3}$),
$^{3}$), and NOx (ml L$^{-1}$). Isopycnals are indicated with a solid white line (contour interval = 0.5 kg m$^{-3}$).

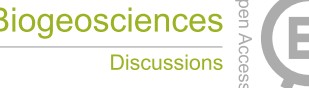


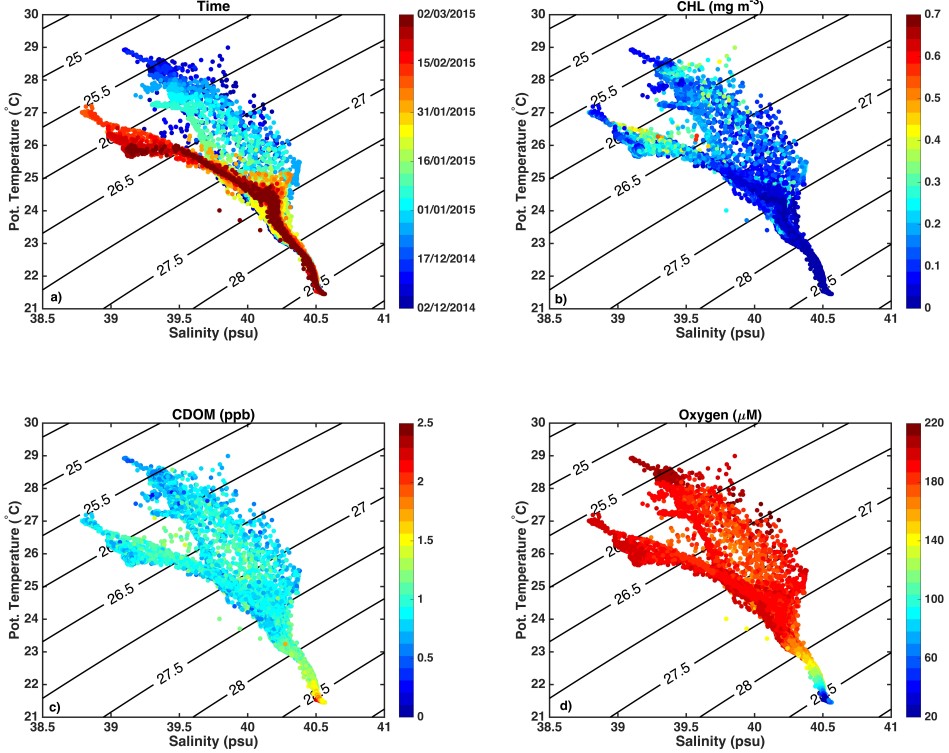


**Figure 5:** Pot. temperature–salinity diagrams from the two glider mission during the winter/early spring
period colored by a) date,  b) CHL (mg m$^{-3}$), c) CDOM (ppb) and d) oxygen (μM).




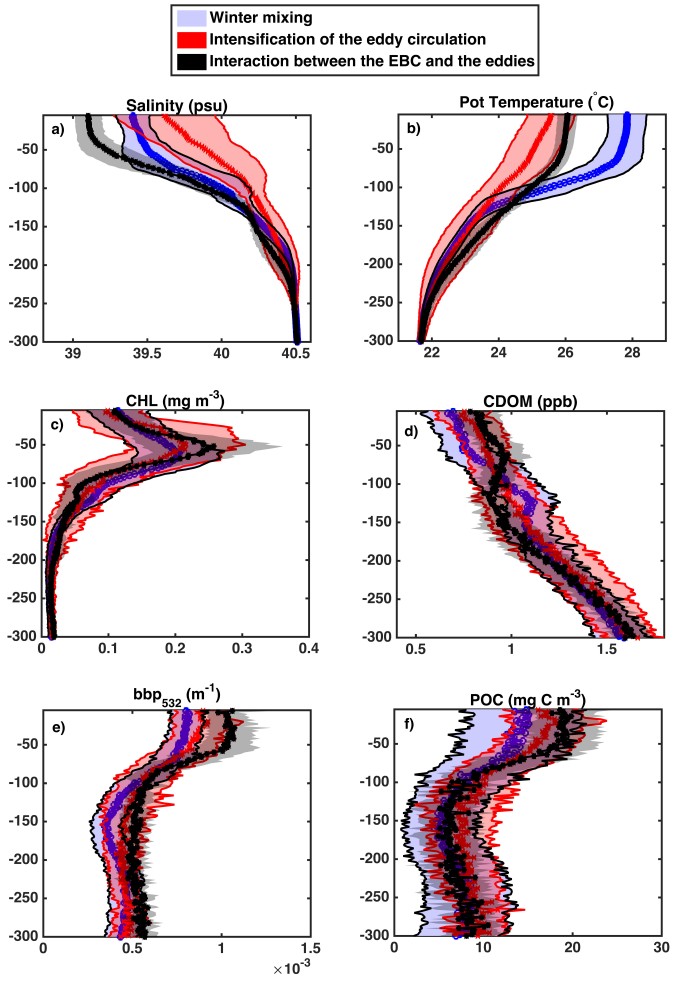


**Figure 6:** Mean vertical profiles of a) salinity (psu), b) pot. temperature (˚C), c) CHL (mg m$^{-3}$) d)
CDOM (ppb) e) bbp$_{532}$ (m$^{-1}$), and f) POC (mg C m$^{-3}$) during the winter mixing (blue line), the
intensification of the eddy circulation (red line) and during the EBC /eddy interaction period (black
line). Shaded areas indicate the standard deviation for each period.

941

942



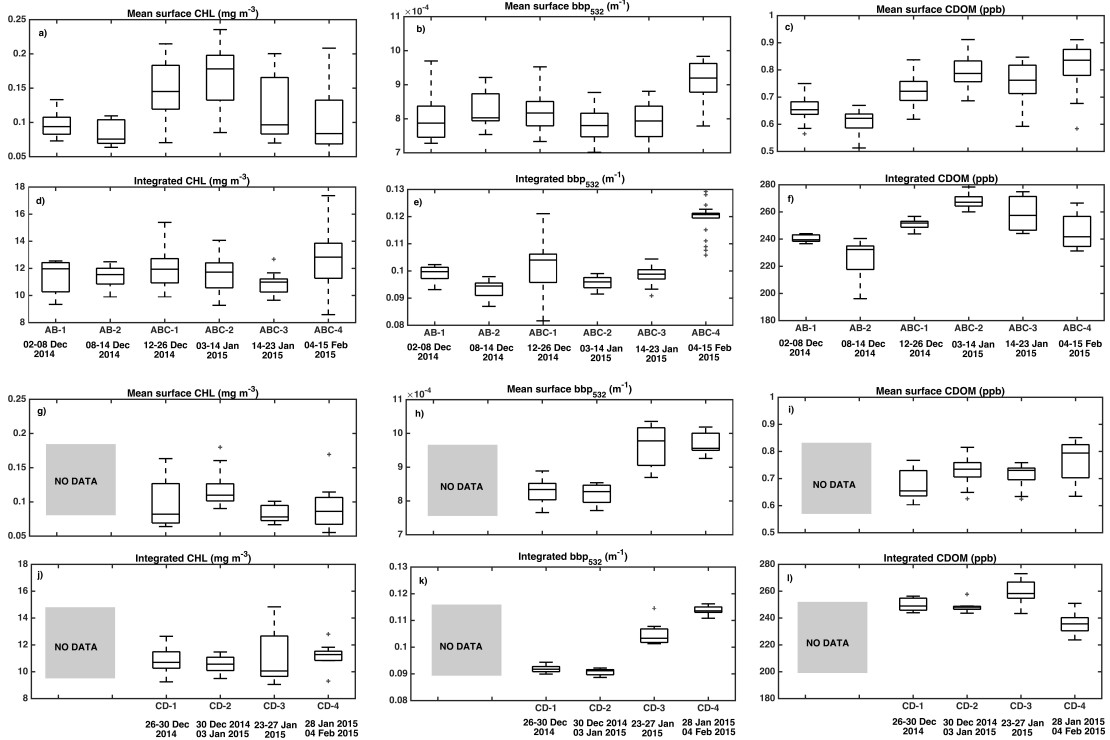

943

**Figure 7:** Box-and-whiskers plots for latitudinal (AB, AC) and longitudinal (CD) transects for the average 0-20m and integrated 0-200m for CHL (mg m$^{-3}$), bbp$_{532}$ (m$^{-1}$), CDOM (ppb) and POC (mg C m$^{-3}$) during the winter/spring period. The mean value of CHL, bbp$_{532}$, CDOM, and POC are shown with the range of variation as well the 25% and 75% quartiles.






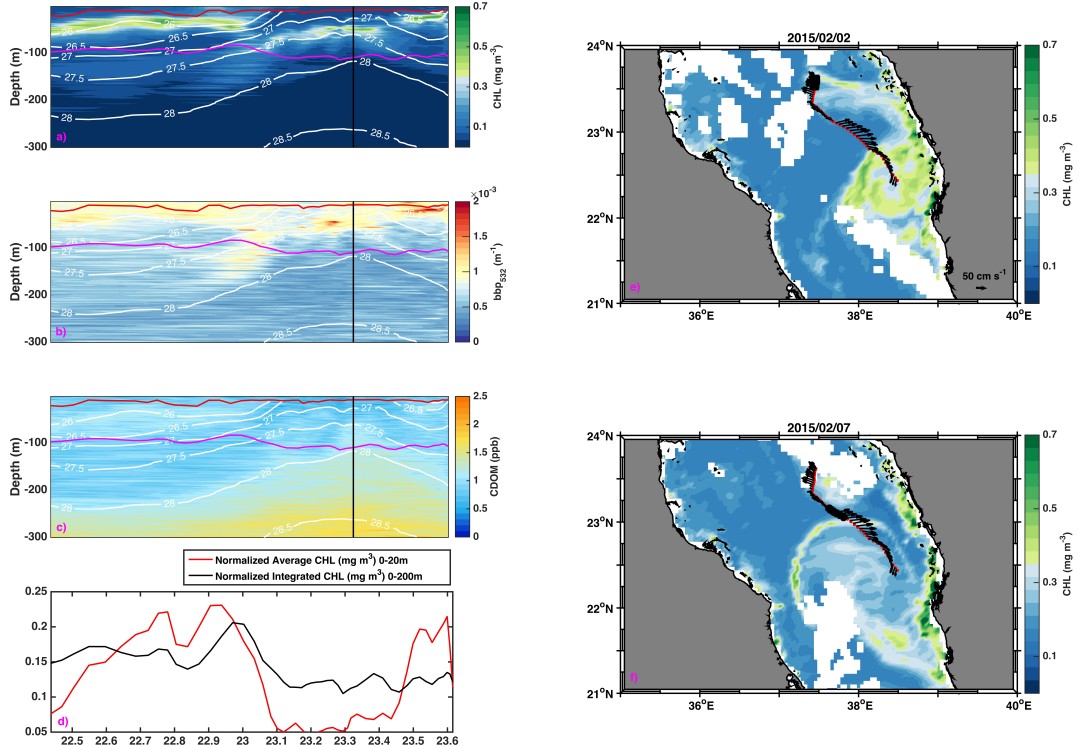


**Figure 8:** Snapshot of the front that developed during the EBC/eddy interaction. Seaglider section plots
for a) CHL b) bbp$_{532}$ and c) POC during the period of 2$^{nd}$ to 11$^{th}$ of February. Normalized 0-20m
average CHL (red line) and 0-200m integrated CHL (black line). Daily MODIS-Aqua CHL for e) 2
February 2015 and f) 7 February 2015. Red dots indicate the glider location and black vectors the glider
derived surface velocity.





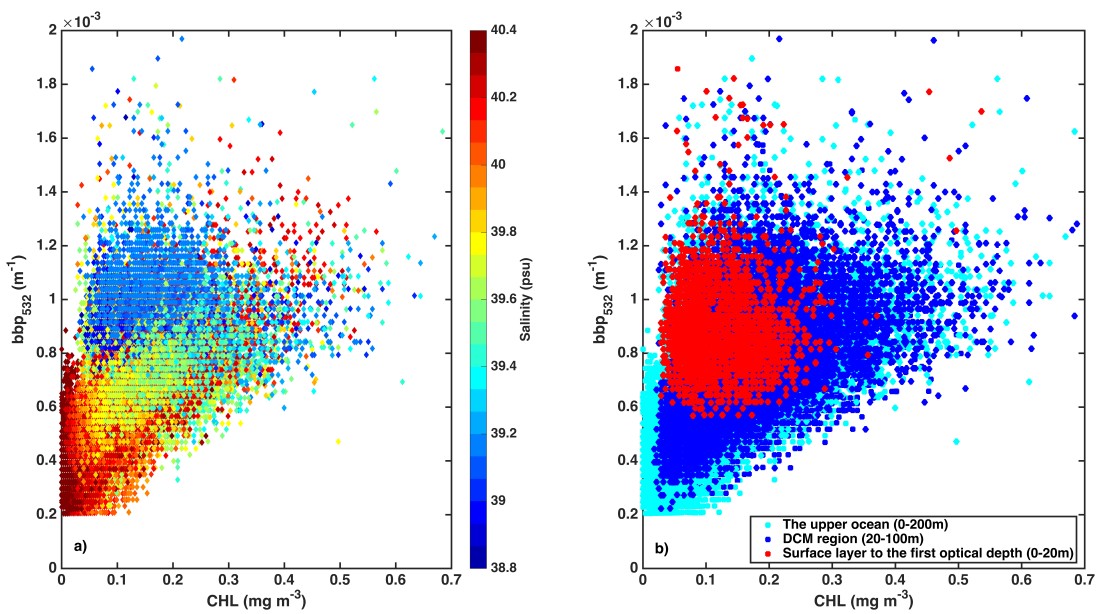


**Figure 9:** Scatterplot of a) bbp$_{532}$ (m$^{-1}$) versus CHL (mg m$^{-3}$) where color indicates the salinity (psu). b)

bbp$_{532}$ (m$^{-1}$) versus CHL (mg m$^{-3}$) in three depth categories: the upper ocean (0-200 m), DCM layer (20-

90 m) and the first optical depth (0-20 m).

960