# Peer review of "Winter mixing, mesoscale eddies and eastern boundary current: Engines for biogeochemical"

_Biogeosciences, 2017_

## Referee Comment (RC1) · Anonymous Referee #1 · 11 Feb 2018

Numerous studies during recent years have focused on the central Red Sea, including on the relationship between physical processes and biological activity (e.g. Chen eta al., 2014; Zarokanellos et al., 2017ab; Triantafyllou et al., 2014; Dreano et al., 2016; Wafar et al., 2016). The present study presents new data, mainly collected by ocean gliders, to investigate the effects of winter mixing, lateral advection and eddies on biological activity. The data is new, but parts of the results are either known or expected (e.g. Zarokanellos et al., 2017ab; Triantafyllou et al., 2014; Dreano et al., 2016; Wafar et al., 2016). As the data is limited to a short period, many of the statements on the

relationship between different variables do not seem robust. In addition, there are no sensitivity tests to some of the arbitrary divisions into wind speed and glider sections.

I would suggest the author to try synthesizing the data and results from the above-motioned references (and few others) instead of focusing on a limited data set that cannot provide conclusive statements (examples below). Altogether, there is now a bulk of data from different sources and different seasons, that can be used to study physical-biological coupling in the Red Sea.

Additional comments: The authors divided the period into three sections: winter mixing, intensified eddy, and strong eastern boundary current. However, no support to this division is provided. Can you demonstrate using satellite data, field observations, or operational numerical simulation the existence of strong eddy and EBC during the corresponding periods? Fig. 8, which is shown only toward the end of the manuscript, provides very limited evidence.

Page 9, line 228: are the results sensitive to the specific division?

Section 3.1: a. Show the lag correlation between air and sea temperature. b. MLD is maximal when Daily mean air temperature is minimal on Jan 11, but similar air temperature appears on Feb 20, and MLD then is rather shallow... c. Overall, weak correlation between wind stress and MLD. How sensitive are the correlations to the division into wind speed categories?

Page 10. Line 273: MLD mixing is not monotonic, and there are cases with strong wind and no mixing.

Page 10, line 276: Where do we see the existence of CE and AE? (the sections shown in Figure 3 are not enough to demonstrate that there are both CE and AE). Also, we expect the MLD to respond differently in CE and in AE.

Section 3.4: can you distinguish between the contribution of the introduction of a new watermass from the Gulf of Aden and uplift of isopycnal by the CE?

Minor comments (partial list) Cetinic′ et al. (2009) Is missing Figure 1: the blue is hardly seen Line 127: extra space Line 142: significantly? Line 207: "study region"? Line 548: there-> that

---

## Referee Comment (RC2) · Anonymous Referee #2 · 20 Feb 2018

This paper deals with physical processes impacting the biogeochemical variability in the central Red Sea using underwater gliders data. These specific platforms are adapted to the studied processes in the study (i.e. meso and sub-mesoscale processes) making relevant data available. Thus, some results of this study could be publishable and valuable for the scientific community in the future. However, the paper needs a lot of work before publication. There are some major issues, especially with the processing of the bio-optical data and their transformation in biogeochemical properties. Moreover, the draft is unfinished with many errors especially in the legends of

the figures, which is unpleasant for the reader and makes the paper very hard to follow. Thus, from my point of view, the current version of the paper is not publishable in the journal Biogeosciences. More details are provided below:

GENERAL COMMENTS

-As stated above, further work on the processing of data has to be done. Major issues on the transformation of raw data in biogeochemical parameters need to be fixed.

-The different sections used to discuss some physical processes (e.g. winter mixing, eddy) are chosen arbitrary from the author. This needs to be supported by more scientific arguments. Moreover, the authors have to provide evidences of the eddy existence (maybe using satellite data?) but also details of the position of the glider from the eddy core during the data acquisition.

-I think that the text has to be re-ordered. The discussion is more a "Results" part (see details in specific comments). The authors could merge the discussion and the results parts in a "Results and Discussion" part.

-This is very hard for the reader to understand the figures of this paper. Some figures are difficult to read, or are even unreadable (e.g. too tiny labels, see specific comments). Moreover, there are too much errors in the legends of the figures for a submitted paper (e.g. parameters not plotted)!

-Authors have to harmonize the abbreviations in the document.

SPECIFIC COMMENTS

-Abstract

l. 21: I think that this statement has to be moderated as the nitrate concentration is not directly measured from the glider.

l. 23: CHL instead of chlorophyll

l. 30: ML instead of mixed layer

-Introduction

l. 68: Please add in the Red Sea.

l. 70-71: Satellites do not sample, I would re-formulate this sentence.

l. 77: CHL has not been introduced before in the text (except in the abstract).

l. 77-78: I do not really agree this sentence... Or at least add references.

l. 91: The authors should do a link between primary production, particle concentration (which one?) and bio-optical properties.

l. 96: Some references should be added here.

l. 100: bbp is the particulate backscattering coefficient and not backscatter.

l. 99-100: Authors do not provide in this study relationships between CHL and CDOM or bbp and CDOM.

l. 102-103: Where this is shown in the paper?

l. 104-105: I would remove this paragraph, that is not essential.

-Data Sources and methodology – Glider observations

l. 119: Where is located the reef system Shi'b Nazar?

l. 131-132: Did the authors remove both negative and positive spikes? If yes, why? Positive spikes have a biological meaning.

l. 146: Even if they not present any measurement for the Red Sea, I think that it is important to mention the paper of Roesler et al. 2017 (Limnology and Oceanography: Methods).

l. 147: CHL measurements: where, when and how these measurements were acquired?

Did the CHL profiles were corrected from the instrumental dark signal?

l. 147-148: I think Morel and Maritorena reference is not well placed because it refers to the calculation of Zeu from Chl?

l. 150 – 183: Are there some differences with the processing of bbp data from profiling floats (see Schmechtig et al. 2014)? And if yes, what are these differences?

l. 173: bbp is commonly referred as $b_{bp}$ (with bp as an index). Moreover, in the equation bbp is well referred but not in the text.

l. 185-186: From my point of view, there are some important issues here.

1) no explications are given for the conversion of bbp(532) in bbp(700), unless it is the reference to Tiwari and Shanmugam, 2013? If yes, the reference is not well placed, the authors should move this reference to the first point of the sentence. As the glider measures the bbp at 3 wavelengths, I would suggest to compute the spectral dependency of the bbp to retrieve the bbp(700). See Loisel et al. 2006 for instance.

2) "and then to POC concentrations using the various empirical relationships". The word "various" is misleading, which one is used here? Or maybe this is an average of all relationships presented in the Table 2 of the paper of Cetinic et al., 2012? Unless I make a mistake, Boss et al. 2013 do not present any measurements of bbp and estimate the POC from the relationship of Gardner et al. 2006. If the authors use the relationship of Cetinic et al. 2012 (which I think is the case because bbp(700) is used) this is a linear relationship so it is not necessary to present both parameters in the figures.

In any case, authors have to moderate the use of POC in their study because many uncertainties are associated with the conversion of bbp in POC, even more in semi-confined environments with specific physical and biogeochemical properties associated, such as the Red Sea.

**BGD**

l. 196: How AOU data were calibrated?

l.199-200: "Because the regression intercepts the oxygen axis between ..", where this statement is shown?

-Data Sources and methodology – Satellite data

Authors have to provide the resolution (spatial + temporal) of the satellite data used in this study.

-Data Sources and methodology – Statistical Analysis

L 225 + l. 228 + L. 231 + l. 238: Tables have to be introduced in the "Results" part.

l. 227: integrated in which layer?

l. 232: The first optical depth has not been introduced before and it is never explained how this is computed?

l. 236-237: Abbreviations are not the same throughout the document.

-Results

As I mentioned above, I think that a lot of work is needed upstream (transformation of bio-optical data in biogeochemical parameters) before reviewing all this "results" part. Thus, here are some comments but it is not an exhaustive list:

l. 256: what means 5m? The authors should show the lag in Figure 2.

l. 258: In Figure 2, the reader cannot see the correlations that the authors mention in the text. Some statistics are needed here.

l. 270 – 280: The authors have to better explain with more details this part of the study. Some data have to support the arbitrary choices of the different sections.

l. 282: How is the NOx computed? It is not well explained in the Material and Methods. Is this a linear relationships based on the AOU? If yes, this plot does not bring any

information to the reader. This is the same issue for the POC with bbp data.

l. 322: How to be sure that this is the core of the eddy?

l. 334: I think that the values for the integrated Chl are false; this is not the range of values expected for the integrated Chl. Do the authors mean the Chl in the whole water-column?

The authors should discuss the ratio Chl:bbp and photoacclimation process.

The Figure 9 does not show any relationships. Where are the statistics associated? See Barbieux et al. 2018 for the red sea. The authors should transform their axis as log/log in order to discuss their results by comparison with the results presented in Barbieux et al., 2018.

-Tale 1

The abbreviations have not been defined before (e.g. what C refers to?).

-Tale 2

It would be useful to add the number of points used to compute each correlation. I am not convinced that the relationships shown here are really robust because of the few data used.

-Table 4

What the grey lines indicate?

-Table 5

Exceed 50% of what?

-Figure 2

The authors have to harmonize sea and air temperature and to add the sea temp in the legend.

-Figure 3

Define the horizontal axis in the legend (latitude?). In the last sentence of the legend, detail in which panel is represented the white line. I would suggest to harmonize the colorbars, except maybe for the BVF, there is no need to have different colorbars for the others parameters. I would put the a), b), etc.. . . in black. The line is blue and not magenta. Labels are too small, it is difficult to read. Is it really necessary to plot the parameters up to 400 m in Figs 2 and 3? I would suggest only 250 m to better see the variability of biogeochemical parameters.

-Figure 4

Same comments as for Figure 3. Moreover, there is no plot with NOx. . . .. The min and max values in the colorbars are not well defined, the reader cannot see correctly the parameters dynamics. The authors shouls limit the plot to 0-250 m as suggested above. I would not put POC panels (linear relationship with bbp).

-Figure 5

Colorbar of panel c not really adapted as every point seem to be green or blue

-Figure 6

POC and bbp: same issue as in Figure 4

-Figure 7

Labels are too small, it makes the plot unreadable. Moreover, the legend does not help the reader as nothing is well explained for guiding the reader (e.g. no reference to panels a), b) . . .)! There is no panel for POC!

-Figure 8

The panels a), b) and c) are already presented in Figure 4. The velocity is not visible in the plot, it is much too small.

-Figure 9

The upper ocean (0-200 m) is in the 2 other categories, do the author mean 100-200 m? If the 2 categories DCM region and Surface layer are right, the blue points represent the 100-200 m layer.

---

## Author Comment (AC1) · 28 Mar 2018

We deeply thank the reviewer for their constructive suggestions. Below we address point by point their comments.

Responses to Reviewer #1 RC1: Numerous studies during recent years have focused on the central Red Sea, including on the relationship between physical processes and biological activity (e.g. Chen eta al., 2014; Zarokanellos et al., 2017ab; Triantafyllou et al., 2014; Dreano et al., 2016; Wafar et al., 2016). The present study presents

new data, mainly collected by ocean gliders, to investigate the effects of winter mixing, lateral advection and eddies on biological activity. The data is new, but parts of the results are either known or expected (e.g. Zarokanellos et al., 2017ab; Triantafyllou et al., 2014; Dreano et al., 2016; Wafar et al., 2016). As the data is limited to a short period, many of the statements on the relationship between different variables do not seem robust. In addition, there are no sensitivity tests to some of the arbitrary divisions into wind speed and glider sections.

AC: We are grateful to the Reviewer #1 for the constructive comments and suggestions. From our perspective, we believe that the results presented in the manuscript are really new and improve significantly our knowledge for the Central Red Sea. First of all, our analysis contains for first time high-resolutions observations during the winter to spring period for the entire Red Sea. Throughout the manuscript we tried to avoid any kind of "speculation" and we focused on the basic patterns revealed by the available observations. Furthermore, in our opinion, the concluding remarks extracted by the synthesis of the datasets used in the study are carefully substantiated and describe in a clear way the important features and processes that occur in a complicated, previously poorly understood region of the Red Sea. However, the reviewer's comments are very insightful. We address these questions below, hopefully in satisfactory manner of to the reviewer.

RC1: I would suggest the author to try synthesizing the data and results from the above motioned references (and few others) instead of focusing on a limited data set that cannot provide conclusive statements (examples below). Altogether, there is now a bulk of data from different sources and different seasons, that can be used to study physical-biological coupling in the Red Sea.

AC: We agree with the reviewer that a comprehensive synthesis of published and existing data is essential. However, the goal of this paper is to focus on the winter/spring transition in central Red Sea which as not been previously described in the literature. We intend to present a synthesis in a future paper.

RC1: Additional comments: The authors divided the period into three sections: winter mixing, intensified eddy, and strong eastern boundary current. However, no support to this division is provided. Can you demonstrate using satellite data, field observations, or operational numerical simulation the existence of strong eddy and EBC during the corresponding periods? Fig. 8, which is shown only toward the end of the manuscript, provides very limited evidence.

AC: Revised as suggested (lines 283-293). We describe the division of periods based on changes in the physical structure observed with the glider during the observational period. We examined the Absolute Dynamic Topography (AVISO) but did not see a clear signature of the eddy despite the strong signal of the glider data. We also examined MODIS ocean color data. The two best images from the study period are shown in figure 8. The CHL patterns are consistent with the surface velocity field measured by the glider and with subsurface density structure. Real time, operational numerical simulations are not yet available for the period.

RC1: Page 9, line 228: are the results sensitive to the specific division?

AC: We followed up with the reviewer's question, which we had not previously done. In fact, the line segment AB is on average 15% greater than the values from the segment ABC. We have modified the table and figure 7 so for all the six glider lines we compare only the AB segment

RC1: Section 3.1: a. Show the lag correlation between air and sea temperature.

AC: We have included a clarification statement in the revised manuscript in the 3.1 section (lines 264-267).

b. MLD is maximal when Daily mean air temperature is minimal on Jan 11, but similar air temperature appears on Feb 20, and MLD then is rather shallow: : :

AC: We have included clarification statements in the revised manuscript in the 3.1 section (lines 268-280)

c. Overall, weak correlation between wind stress and MLD. How sensitive are the correlations to the division into wind speed categories?

AC: We are grateful for the reviewer recommendation and we removed this part of the analyses as the overall correlation between wind stress and MLD is weak. Revised as suggested (lines 277-280).

Page 10. Line 273: MLD mixing is not monotonic, and there are cases with strong wind and no mixing.

AC: Revised as suggested (lines 269-277).

RC1: Page 10, line 276: Where do we see the existence of CE and AE? (the sections shown in Figure 3 are not enough to demonstrate that there are both CE and AE). Also, we expect the MLD to respond differently in CE and in AE.

AC: We thank the reviewer for his comment and we agree that the overall AE is not well defined from the existing observations. Revised as suggested (lines: 287-290; 303-304, 307, 311-314, 320-323, 326-327, 331-333 338-339, 358-361, 372-375,393-394, 534-535 and 647)

RC1: Section 3.4: can you distinguish between the contribution of the introduction of a new watermass from the Gulf of Aden and uplift of isopycnal by the CE?

Of course all water in the Red Sea originates from the Gulf of Aden. We can distinguish the water mass that has most recently originated from the Gulf of Aden based on thermohaline and optical properties [Churchill et al., 2014; Zarokanellos et al., 2017b]. CDOM is not typically used as water mass tracer but we have found that is consistently indicative of water that has recently originated from Gulf of Aden. Recent studies from other regions have shown the utility of CDOM for tracing and differentiating water with unique characteristics [e.g. Seegers et al., ECSS, 2016].

RC1: Minor comments (partial list) Cetinic et al. (2009) Is missing

AC: Revised as suggested (lines 736-738)

RC1: Figure 1: the blue is hardly seen

AC: Revised as suggested (Figure 1 has been updated)

RC1: Line 127: extra space

AC: Revised as suggested

RC1: Line 142: significantly?

AC: Revised as suggested (line 145)

RC1: Line 207: "study region"?

AC: Revised as suggested (line 221)

RC1: Line 548: there-> that

AC: Revised as suggested (line 570)
* * *
[Figure]

**Fig. 1.** Revised Figure 1

---

## Author Comment (AC2) · 12 Apr 2018

Responses to Reviewer #2

RC2: This paper deals with physical processes impacting the biogeochemical variability in the central Red Sea using underwater gliders data. These specific platforms are adapted to the studied processes in the study (i.e. meso and sub-mesoscale processes) making relevant data available. Thus, some results of this study could be publishable and valuable for the scientific community in the future. However, the pa-

per needs a lot of work before publication. There are some major issues, especially with the processing of the bio-optical data and their transformation in biogeochemical properties.

RC2: Moreover, the draft is unfinished with many errors especially in the legends of the figures, which is unpleasant for the reader and makes the paper very hard to follow. Thus, from my point of view, the current version of the paper is not publishable in the Journal Biogeosciences. More details are provided below:

AC: We thank the reviewer for the constructive comments and suggestions. We address his/her comments below.

GENERAL COMMENTS RC2: As stated above, further work on the processing of data has to be done. Major issues on the transformation of raw data in biogeochemical parameters need to be fixed.

RC2: The different sections used to discuss some physical processes (e.g. winter mixing, eddy) are chosen arbitrary from the author. This needs to be supported by more scientific arguments. Moreover, the authors have to provide evidences of the eddy existence (maybe using satellite data?) but also details of the position of the glider from the eddy core during the data acquisition.

AC: We describe the division of periods based on changes in the physical structure observed with the glider during the observational period. We examined the Absolute Dynamic Topography (AVISO) but did not see a clear signature of the eddy despite the strong signal of the glider data. We also examined MODIS ocean color data. The two best images from the study period are shown in figure 8. The CHL patterns are consistent with the surface velocity field measured by the glider and with subsurface density structure. Real-time, operational numerical simulations are not yet available for the period. We have revised the manuscript to:" Based on the physical structure that was observed during the winter/early spring period, we have defined three distinct periods: a) the first period, from early December through mid-January, is characterized

by several deeper mixing events to more than 50m as shown in Figure 2d. Deepening of the ML is typically associated with increased wind stress, but not every increase of wind stress yielded deepening of the ML (Fig. 2b and 2d); b) the second period, from 11 January through 10 February 2015, was characterized by intensification of the cyclonic eddy (CE) circulation when the 27.5 kg m-3 isopycnal ascended from 135m in late December to 30m in mid-January. During this period, relative humidity and air temperature gradually increased, and the ML became progressively shallower; and c) the last period when the CE is still evident and warmer and less saline water in the upper layer suggest advection of water from the south into the region. This water was first detected when the glider was southbound along the transect shown in Figure 3i and 3j". Lines 283-293 of the revised manuscript.

RC2: I think that the text has to be re-ordered. The discussion is more a "Results" part (see details in specific comments). The authors could merge the discussion and the results parts in a "Results and Discussion" part.

AC: We combined the result and discussion section as suggested. We have also re-structured the subsections 3.7 and 3.8 to be more clearly delineated.

RC2: This is very hard for the reader to understand the figures of this paper. Some figures are difficult to read, or are even unreadable (e.g. too tiny labels, see specific comments).

AC: Figures revised as suggested

RC2: Moreover, there are too much errors in the legends of the figures for a submitted paper (e.g. parameters not plotted)!

AC: Legends revised as suggested.

RC2: Authors have to harmonize the abbreviations in the document.

AC: Revised as suggested

SPECIFIC COMMENTS -Abstract RC2: l. 21: I think that this statement has to be moderated as the nitrate concentration is not directly measured from the glider.

AC: The reviewer brings up an important point and we have clarified the abstract in the revised manuscript to: "During winter and early spring, surface cooling and stronger winds resulted in deepening of the mixed layer (ML) up to 90m. Based on the distribution of density and oxygen it appears that the ML did not penetrate into the nutricline. However, the mixing events dispersed phytoplankton from the deep CHL maximum throughout the ML increasing the chlorophyll signature detected by ocean colour imagery. In early spring, low salinity water that originates in the Gulf of Aden began to appear in the CRS. Relatively high concentrations of CHL and CDOM, suggestive of previous nutrient availability, along with lower oxygen concentrations are associated with this low salinity water". Lines 20-22 and 24-27 of the revised manuscript

RC2: l. 23: CHL instead of chlorophyll

AC Revised as suggested (line 23)

RC2: l. 30: ML instead of mixed layer

AC: Revised as suggested (line 32)

RC2: Introduction l. 68: Please add in the Red Sea.

AC: Revised as suggested (line 71-73)

RC2: l. 70-71: Satellites do not sample, I would re-formulate this sentence.

AC: We have revised the manuscript to: "Until now, only remote sensing studies have been utilized to examine the seasonal variability of phytoplankton (Acker et al., 2008a; Racault et al., 2015; Raitsos et al., 2013b). Despite remote sensing's ability to provide synoptic imagery of key ocean variables, persistent clouds, atmospheric aerosols, sunglint, and sensor saturation over land limit data acquisition for the Red Sea (Racault et al., 2015). However, satellite imagery suggests that there is an ecological connection

between the coastal zone and the central open sea. " Lines 73-78 of the revised manuscript.

RC2: l. 77: CHL has not been introduced before in the text (except in the abstract).

AC: Revised as suggested (line 82)

RC2: l. 77-78: I do not really agree this sentence: : : Or at least add references.

AC: We have clarified this sentence in the revised manuscript to: "The ocean color measurements often give us evidence of the water exchange between coastal coral reefs and the open sea (Raitsos et al., 2017)". Revised as suggested (lines 79-80)

RC2: l. 91: The authors should do a link between primary production, particle concentration (which one?) and bio-optical properties.

AC: We have revised the manuscript to:. "Lateral transport and eddy stirring can contribute to the carbon flux in a given region (McGillicuddy, 2016). The distributions of biogeochemical variables including CHL, suspended particles, nutrients and oxygen enable us to understand the coupling between various scales of physical forcing and variability with biogeochemical dynamics (e.g Mahadevan et al., 2012; Mignot et al., 2014; Perry et al., 2008). The use of bio-optical variables from both remote sensing and in-situ observations has begun to yield new insights into the biogeochemical functioning of Red Sea (Acker et al., 2008b; Brewin et al., 2015; Dreano et al., 2015; Gittings et al., 2018; Kheireddine et al., 2018; Kürten et al., 2016; Racault et al., 2015; Raitsos et al., 2013a; Tiwari et al., 2018; Zarokanellos et al., 2017b, 2017a). Lines 94-102 of the revised manuscript.

RC2: l. 96: Some references should be added here.

AC: We have revised the manuscript to: "In a phytoplankton dominated regime a number of optical properties will covary with CHL concentration (Bricaud et al., 1988; Siegel et al., 2005), but other constituents are capable of absorbing and scattering light, such as detrital particles, CDOM, and suspended solids, which all contribute independently

to ocean color (Kheireddine et al., 2018; Loisel et al., 2002, 2007; Maritorena and Siegel, 2005; Tiwari et al., 2018)". This is a representative but not an exhaustive list of references. Lines 103-107 of the revised manuscript.

RC2: l. 100: bbp is the particulate backscattering coefficient and not backscatter.

AC: Revised as suggested (line 110)

RC2: l. 99-100: Authors do not provide in this study relationships between CHL and CDOM or bbp and CDOM.

AC: We have revised the manuscript to: Additionally, the relationship between CHL and particulate backscattering coefficient at 532nm (bbp532) has been examined and provide information on the primary production and optical variability in time and space". Lines 110-112 of the revised manuscript.

RC2: l. 102-103: Where this is shown in the paper?

AC: This line has been removed.

RC2: l. 104-105: I would remove this paragraph, that is not essential.

AC: Revised as suggested

RC2: Data Sources and methodology – Glider observations RC2: l. 119: Where is located the reef system Shi'b Nazar?

AC: We have revised the manuscript to:" Each deployment began offshore of a reef system Shi'b Nazar (located at 22ËŽ16'12 ÎÌ, 38ËŽ 57'44 E; reef not shown in Fig 1b) and the glider was piloted northwestward to offshore turning point where it turned northeastward toward the northern end point near Yanbu. The glider then returned to southern waypoint retracing the same track and completing the round trip in about three weeks". Lines 121-125 of the revised manuscript

RC2: l. 131-132: Did the authors remove both negative and positive spikes? If yes,

why? Positive spikes have a biological meaning.

AC: We have revised the manuscript to:" Spikes from the raw hydrographic measurements have been removed. In addition, CTD data have been examined for thermal lag effect and no evidence of its effect has been observed. FL3 and BB3 sensors were manufacturer calibrated. Negative spikes and instrument dark signal have been removed from the fluorescence profiles." Revised as suggested (line 148-152)

RC2: l. 146: Even if they not present any measurement for the Red Sea, I think that it is important to mention the paper of Roesler et al. 2017 (Limnology and Oceanography: Methods).

AC: We have revised the manuscript to: " In addition, a global bias correction factor of two has been applied for the fluorescence profiles. This correction is based on the comparison of fluorometer and extracted CHL measurements (Roesler et al., 2017)." Revised as suggested (line 152-154)

RC2: l. 147: CHL measurements: where, when and how these measurements were acquired?

AC: We have revised the manuscript to: "In addition, a global bias correction factor of two has been applied for the fluorescence profiles. This correction is based on comparison of fluorometer and extracted CHL measurements (Roesler et al., 2017)."Revised as suggested (line 152-154)

RC2: Did the CHL profiles were corrected from the instrumental dark signal?

AC: Yes we did that. The manuscript revised as suggested (line 151-152).

RC2: l. 147-148: I think Morel and Maritorena reference is not well placed because it refers to the calculation of Zeu from Chl?

AC: This line has been removed, as we do not use the Zeu calculation.

RC2: l. 150 – 183: Are there some differences with the processing of bbp data from

profiling floats (see Schmechtig et al. 2014)? And if yes, what are these differences?

AC: There are some differences when you processing bbp data from the floats and glider and primarily depends on the sensor that the platform is equipped. Until now, there are 7 sensors available in the market and some of them they have different centroid angle and x value that you need to take into your consideration when you process your bbp data.

RC2: l. 173: bbp is commonly referred as b[bp] (with bp as an index). Moreover, in the equation bbp is well referred but not in the text.

AC: Revised as suggested

RC2: l. 185-186: From my point of view, there are some important issues here. 1) no explications are given for the conversion of bbp(532) in bbp(700), unless it is the reference to Tiwari and Shanmugam, 2013? If yes, the reference is not well placed, the authors should move this reference to the first point of the sentence. As the glider measures the bbp at 3 wavelengths, I would suggest to compute the spectral dependency of the bbp to retrieve the bbp(700). See Loisel et al. 2006 for instance. 2) "and then to POC concentrations using the various empirical relationships". The word "various" is misleading, which one is used here? Or maybe this is an average of all relationships presented in the Table 2 of the paper of Cetinic et al., 2012? Unless I make a mistake, Boss et al. 2013 do not present any measurements of bbp and estimate the POC from the relationship of Gardner et al. 2006. If the authors use the relationship of Cetinic et al. 2012 (which I think is the case because bbp(700) is used) this is a linear relationship so it is not necessary to present both parameters in the figures.

AC: We have revised the manuscript to: "The spectral dependence ($\gamma$) of particulate backscattering coefficient has been estimated using the wavelengths at 532, 650 and 880 nm (Loisel et al., 2006). The spectra slope ($\gamma$) defines the shape, magnitude, and the variability of the particulate backscattering spectra. The $\gamma$ values are in agreement with the study of Loisel et al. (2016) that low chlorophyll waters are associated with

small-sized particles and high $\gamma$ values (mean= 1.7214; STD $\pm$ 0.97). Then, using the power law model of the particulate backscattering dependency we transform b_bp532 to b_bp700 (Loisel et al., 2006; Tiwari and Shanmugam, 2013).

The b_bp700 can be defined as: b_bp700=b_bp532$\times$(700/532)$\hat{}\gamma$

where b_bp700and b_bp532 are the particulate backscattering coefficient at the desired and the reference wavelengths, respectively. POC concentrations have been defined from the empirical relationship of the b_bp700 with POC (Cetinić et al., 2012)." Lines 193-205 of the revised manuscript.

RC2: In any case, authors have to moderate the use of POC in their study because many uncertainties are associated with the conversion of bbp in POC, even more in semi-confined environments with specific physical and biogeochemical properties associated, such as the Red Sea.

AC: Revised as suggested

RC2: l. 196: How AOU data were calibrated?

AC: We have revised the manuscript to: "The relationship between apparent oxygen utilization (AOU) and nitrate was determined using two-way linear regression analysis (Naqvi et al., 1986). From the relationship of salinity, temperature, and oxygen concentration from our glider deployments, we determined dissolved nitrate plus nitrite (NOx) based on apparent oxygen utilization (see Churchill et al., 2014). Recent evaluations of the relationship between AOU and NOx have shown a consistent relationship (Churchill et al., 2014; Naqvi et al., 1986; Zarokanellos et al., 2017b). Because the regression intercepts the oxygen axis between 180-185 $\mu$M (not shown here), we use the 180 $\mu$M oxygen isopleth as an index of the nitracline depth to examine the nutrient availability in the euphotic layer." Revised as suggested (lines 208-215)

RC2: l.199-200: "Because the regression intercepts the oxygen axis between ..", where this statement is shown?

AC: We have revised the manuscript to: "The relationship between apparent oxygen utilization (AOU) and nitrate was determined using two-way linear regression analysis (Naqvi et al., 1986). From the relationship of salinity, temperature, and oxygen concentration from our glider deployments, we determined dissolved nitrate plus nitrite (NOx) based on apparent oxygen utilization (see Churchill et al., 2014). Recent evaluations of the relationship between AOU and NOx have shown a consistent relationship (Churchill et al., 2014; Naqvi et al., 1986; Zarokanellos et al., 2017b). Because the regression intercepts the oxygen axis between 180-185 $\mu$M (not shown here), we use the 180 $\mu$M oxygen isopleth as an index of the nitracline depth to examine the nutrient availability in the euphotic layer." Revised as suggested (line 208-215)

RC2: Data Sources and methodology – Satellite data Authors have to provide the resolution (spatial + temporal) of the satellite data used in this study.

AC: We have revised the manuscript to: "To place the glider observations into the larger context of the CRS two relatively clear daily MODIS Level-3 data set with 4 km horizontal resolution were obtained from NASA's Ocean Color Web (http://oceancolor.gsfc.nasa.gov/) for 2 and 7 February 2015." Revised as suggested (lines 233-235)

RC2: Data Sources and methodology – Statistical Analysis L 225 + l. 228 + L. 231 + l. 238: Tables have to be introduced in the "Results" part.

AC: Revised as suggested

RC2: l. 227: integrated in which layer?

AC: Revised as suggested (lines 238-240). We refer to the integrated CHL, CDOM, bbp532, and POC in the ML.

RC2: l. 232: The first optical depth has not been introduced before and it is never explained how this is computed?

AC: We have revised the manuscript to:" In addition, mean values of CHL, bbp532,

cyclonic eddy (CE) circulation when the 27.5 kg m-3 isopycnal ascended from 135m in late December to 30m in mid-January. During this period, relative humidity and air temperature gradually increased, and the ML became progressively shallower; and c) the last period when the CE is still evident and warmer and less saline water in the upper layer suggest advection of water from the south into the region. This water was first detected when the glider was southbound along the transect shown in Figure 3i and 3j. The glider time series shows the evolution of salinity, potential temperature, stratification (Brunt–VaÍLisaÍLlaÍL Frequency; BVF) and oxygen structure in the upper 300 m during the winter/early spring period (Fig. 3). The in situ observations illustrate the importance of winter mixing, the physical and chemical characteristics of the CE, and the interaction between the eddy with warm, low salinity intrusion." Lines 283-298 of the revised manuscripts.

RC2: l. 282: How is the NOx computed? It is not well explained in the Material and Methods. Is this a linear relationships based on the AOU? If yes, this plot does not bring any information to the reader. This is the same issue for the POC with bbp data.

AC: We have revised the manuscript to: "The relationship between apparent oxygen utilization (AOU) and nitrate was determined using two-way linear regression analysis (Naqvi et al., 1986). From the relationship of salinity, temperature, and oxygen concentration from our glider deployments, we determined dissolved nitrate plus nitrite (NOx) based on apparent oxygen utilization (see Churchill et al., 2014). Recent evaluations of the relationship between AOU and NOx have shown a consistent relationship (Churchill et al., 2014; Naqvi et al., 1986; Zarokanellos et al., 2017b). Because the regression intercepts the oxygen axis between 180-185 $\mu$M (not shown here), we use the 180 $\mu$M oxygen isopleth as an index of the nitracline depth to examine the nutrient availability in the euphotic layer. " Lines 208-215 of the revised manuscript. Figure 4 has been revised.

RC2: l. 322: How to be sure that this is the core of the eddy? AC: The tilting of the isopycnal stimulate biological production through vertical nutrient injection. These nutrient injections create substantial patchiness in biological fields and spatial separation between production and biomass.

RC2: l. 334: I think that the values for the integrated Chl are false; this is not the range of values expected for the integrated Chl. Do the authors mean the Chl in the whole water-column? AC: We have revised the manuscript to: "Mean CHL concentration in the ML ($0.16 \pm 0.03$) during the observation period was higher than the mean CHL ($0.099 \pm 0.015$; 0-200m)". Revised as suggested (lines 351-352).

RC2: The authors should discuss the ratio Chl:bbp and photoacclimation process.

AC: We have revised the manuscript to: "CHL is often an imperfect index of phytoplankton biomass as it can be influenced not only by phytoplankton biomass but also by several environmental conditions that include light availability and exposure, temperature, and nutrient availability (Babin et al., 1996; Barbieux et al., 2018). Phytoplankton can change their intracellular CHL in response to changes of light condition via photoacclimation (Dubinsky and Stambler, 2009; Falkowski and Laroche, 1991). A result of photoacclimation is changed in the bbp:CHL ratio (Cullen, 2015; Siegel et al., 2005).Changes in the composition and size of particles may cause large variability in bbp (Flory et al., 2004)." Lines 574-580 of the revised manuscript

The Figure 9 does not show any relationships. Where are the statistics associated? See Barbieux et al. 2018 for the red sea. The authors should transform their axis as log/log in order to discuss their results by comparison with the results presented in Barbieux et al., 2018.

AC: We have revised the manuscript to: "Figure 9b illustrates the same relationship but color-coded into two categories: a) surface layer to 20m (approximately the first optical depth, or the penetration depth (pd);(Morel and Berthon, 1989)) ) and b) 0 - 1.5 Zeu (120 m; where Zeu is the depth of the euphotic layer as defined by Morel and Berthon (1989). Barbieux et all, 2018 define 1.5 Zeu as the "enlightened layer". Our distribution of CHL and bbp532 shows similar patterns (R2=0.166 and slope=0.238; exponent of

the power law). The variability of the bbp:CHL relationship depends on the region of the water column considered. Within the nearsurface layer that we have used as an approximation of the penetration depth, the relationship shows no clear relationship but rather appears as a centroid around which most of the values fall. Higher values of bbp532 occurred within the shallow depth range (0-20m). CHL values were generally low. Because we do not see significant diel variability in the ML we conclude that this CHL fluorescence variability is not due to quenching." Revised as suggest (lines 586-596)

RC2: Table 1 The abbreviations have not been defined before (e.g. what C refers to?). AC: Revised as suggested (Table 1)

RC2: Table 2 It would be useful to add the number of points used to compute each correlation. I am not convinced that the relationships shown here are really robust because of the few data used.

AC: Based on the suggestion of the first reviewer we remove Table 2 and revised the manuscript. However, for your information, the number of points that we used to compute the correlation was 91.

RC2: Table 4 What the grey lines indicate?

AC: Revised as suggested (Table 3)

RC2: Table 5 Exceed 50% of what?

AC: Grey cells highlight the values that exceed 50% of the mean value. Revised as suggested (Table 4)

RC2: Figure 2 The authors have to harmonize sea and air temperature and to add the sea temp in the legend. AC: Revised as suggested (Update Figure 2)

RC2: Figure 3 Define the horizontal axis in the legend (latitude?). In the last sentence of the legend, detail in which panel is represented the white line. I would suggest to

harmonize the colorbars, except maybe for the BVF, there is no need to have different colorbars for the others parameters. I would put the a), b), etc.: : : in black. The line is blue and not magenta. Labels are too small, it is difficult to read. Is it really necessary to plot the parameters up to 400 m in Figs 2 and 3? I would suggest only 250 m to better see the variability of biogeochemical parameters.

AC: Revised as suggested. However, we keep the different colorbars from each variable as we think that is more suitable for the data visualization.

RC2: Figure 4 Same comments as for Figure 3. Moreover, there is no plot with NOx: : :.. The min and max values in the colorbars are not well defined, the reader cannot see correctly the parameters dynamics. The authors should limit the plot to 0-250 m as suggested above. I would not put POC panels (linear relationship with bbp).

AC: Figure 4 (Update Figure 4) and manuscript revised as suggested.

RC2: Figure 5 Colorbar of panel c not really adapted as every point seem to be green or blue

AC: Revised as suggested ((Update Figure 5)

RC2: Figure 6 POC and bbp: same issue as in Figure 4

AC: Revised as suggested (Update Figure 6)

RC2: Figure 7 Labels are too small, it makes the plot unreadable. Moreover, the legend does not help the reader as nothing is well explained for guiding the reader (e.g. no reference to panels a), b) : : :)! There is no panel for POC!

AC: Revised as suggested

RC2: Figure 8 The panels a), b) and c) are already presented in Figure 4. The velocity is not visible in the plot, it is much too small.

AC: Figure 8 has been updated. We have revised the manuscript to: "The in-situ

observations from this period, reveal subduction of CHL and elevated concentration of CHL at the interface of CE with the low salinity intrusion (Fig. 4i and 4j). The integrated CHL concentration is maximum at their interface where the subductive CHL of the CE and shallower DCM of the freshwater intrusion overlap (22.9 ËŽN; Fig. 8c). The CE accumulate low salinity and relatively high in CHL water in the periphery of the eddy and as result elevate normalize average and integrated CHL has been observed (Fig. 8c)." Lines 529-534 of the revised manuscript.

RC2: Figure 9 The upper ocean (0-200 m) is in the 2 other categories, do the author mean 100- 200 m? If the 2 categories DCM region and Surface layer are right, the blue points represent the 100-200 m layer.

AC: Revised as suggested (updated Figure 9). The manuscript has also been revised (Lines 574-580 and 586-596)
* * *
[Figure]

**Fig. 1.** Revised Figure 2

[Figure]

**Fig. 2.** Revised Figure 3

[Figure]

Winter mixing
2014/12/17 - 2014/12/26

Intesification of the
eddy circulation
2015/01/14 - 2015/01/23

Interaction between the eddy
with warm, low salinity intrusion
2015/02/04 - 2015/02/15

**Fig. 3.** Revised Figure 4

**Fig. 4.** Revised Figure 5

[Figure]

**Fig. 5.** Revised Figure 6

[Figure]

**Fig. 6.** Revised Figure 7

[Figure]

**Fig. 7.** Revised Figure 8

[Figure]

**Fig. 8.** Revised Figure 9